

# Lifshitz symmetry: Lie algebras, spacetimes and particles

José Figueroa-O'Farrill[*], Ross Grassie[†] and Stefan Prohazka[‡]

Maxwell Institute and School of Mathematics, The University of Edinburgh, James Clerk Maxwell Building, Peter Guthrie Tait Road, Edinburgh EH9 3FD, Scotland, United Kingdom

[*] j.m.figueroa@ed.ac.uk , [†] rgrassie@ed.ac.uk , [‡] stefan.prohazka@ed.ac.uk

## Abstract

We study and classify Lie algebras, homogeneous spacetimes and coadjoint orbits ("particles") of Lie groups generated by spatial rotations, temporal and spatial translations and an additional scalar generator. As a first step we classify Lie algebras of this type in arbitrary dimension. Among them is the prototypical Lifshitz algebra, which motivates this work and the name "Lifshitz Lie algebras". We classify homogeneous spacetimes of Lifshitz Lie groups. Depending on the interpretation of the additional scalar generator, these spacetimes fall into three classes:

1. $(d + 2)$-dimensional Lifshitz spacetimes which have one additional holographic direction;

2. $(d + 1)$-dimensional Lifshitz–Weyl spacetimes which can be seen as the boundary geometry of the spacetimes in (1) and where the scalar generator is interpreted as an anisotropic dilation;

3. and $(d + 1)$-dimensional aristotelian spacetimes with one scalar charge, including exotic fracton-like symmetries that generalise multipole algebras.

We also classify the possible central extensions of Lifshitz Lie algebras and we discuss the homogeneous symplectic manifolds of Lifshitz Lie groups in terms of coadjoint orbits.

# 1  Introduction

Since the 1980s, numerous examples of condensed matter systems without a quasiparticle description have been found [1,2]. These systems cannot be described using traditional Landau-Fermi liquid theory; therefore, these non-Fermi liquids, such as cuprate superconductors [3–5], heavy fermion systems near a quantum phase transition [6–8] and graphene in metallic states [9–11] require a new set of tools for understanding their thermodynamic and transport properties [12,13]. In particular, there is great interest in understanding the universal behaviour of certain physical properties near quantum critical points [13–15]. For example, in [16–18], it was shown that resistivity grows linearly with temperature in heavy fermion materials. For systems with Lifshitz invariance, Lorentz invariance is broken, and we observe an anisotropic scaling between time and space,

$$t \to \lambda^z t \quad \text{and} \quad \boldsymbol{x} \to \lambda \boldsymbol{x}, \tag{1}$$

where $z \neq 1$. However, away from these critical points, there is the option that Lorentz invariance is restored, corresponding to $z = 1$ above.

Over the last two decades, holography has developed into a novel tool for investigating the physical properties of condensed matter systems without a quasiparticle description. A wealth of literature has appeared under the title AdS/CMT, which seeks to build a holographic dictionary between the strongly-coupled quantum field theories underlying these systems and gravitational theories (see, e.g., [1,19–22]). This work has led to exciting connections between black hole physics and the thermal physics of strongly interacting condensed matter [22], new ways (see, e.g., [23]) of calculating entanglement entropy in Lifshitz field theories [24,25] and searching for universal properties of non-relativistic field theories [19]; it has also led to fields of research such as holographic superconductors [26] and holographic hydrodynamics [27].

Tailoring this AdS/CMT toolkit to the strongly-coupled systems near criticality mentioned previously, we arrive at a field of research known as Lifshitz holography, so-called due to the anisotropic Lifshitz scaling present in the field theory at the boundary. The gravitational theories dual to these Lifshitz field theories are built upon Lifshitz spacetimes, where the scaling symmetry of the boundary field theory is geometrised as the presence of the radial dimension [19,28]. This transformation of the scaling symmetry into a radial direction has a nice interpretation, which was highlighted in [29]. Namely the Lie algebra of Killing vector fields (KVFs) of the bulk geometry, which is isomorphic to the Lifshitz algebra, becomes the Lie algebra of conformal Killing vector fields (CKVFs) of the boundary geometry. We can view this transformation from KVFs to CKVFs as an artefact of the boundary geometry being the quotient of the bulk geometry with respect to the Killing vector field corresponding to the radial direction.

To see this transformation, consider the $(d + 2)$-dimensional Lifshitz metric [28] (with unit radius of curvature), with exponent $z$, given in local coordinates $(r, t, \boldsymbol{x}) \in (0, \infty) \times \mathbb{R} \times \mathbb{R}^d$ by

$$g = -\frac{dt^2}{r^{2z}} + \frac{dr^2}{r^2} + \frac{d\boldsymbol{x}^2}{r^2}. \tag{2}$$

Here, $d\mathbf{x}^2$ is the euclidean metric on $\mathbb{R}^d$. For generic values of the exponent $z$, the Lie algebra of Killing vector fields of the above metric has dimension $d(d+1)/2 + 2$ and is spanned by

$$\xi_{J_{ab}} = -x_a \partial_b + x_b \partial_a\,, \qquad \xi_{P_a} = \partial_a\,, \qquad \xi_H = \partial_t \qquad \text{and} \qquad \xi_D = r\partial_r + x^a \partial_a + zt\partial_t\,, \quad (3)$$

which satisfy the (opposite) Lie brackets to those of the *Lifshitz Lie algebra* spanned by $J_{ab} = -J_{ba}$, $P_a$, $H$ and $D$:

$$
\begin{aligned}
[J_{ab}, J_{cd}] &= \delta_{bc} J_{ad} - \delta_{ac} J_{bd} - \delta_{bd} J_{ac} + \delta_{ad} J_{bc}\,, \\
[J_{ab}, P_c] &= \delta_{bc} P_a - \delta_{ac} P_b\,, \\
[D, P_a] &= P_a\,, \\
[D, H] &= zH\,,
\end{aligned}
\qquad (4)
$$

with no other nonzero Lie brackets. It is clear that the Killing vector fields above span every tangent space, so that the Lie algebra is transitive and hence the Lifshitz metric is (locally) homogeneous. In fact, it is not hard to show that the spacetime is homogeneous. There are two special values of the exponent for which the metric admits additional Killing vector fields. As mentioned previously, if $z = 1$, then the Lorentz boost invariance of the metric is restored and (2) describes the Poincaré patch of $\mathsf{AdS}_{d+2}$. In this case, the space is maximally symmetric, admitting a Lie algebra of isometries of dimension $(d+2)(d+3)/2$. The other special value is $z = 0$. With this value for the exponent, equation (2) describes a conformally flat lorentzian metric on the product of a timelike line with $(d+1)$-dimensional hyperbolic space.

Notice that the Killing vector $\xi_D$ is the only vector field with a $\partial_r$ component. The boundary at $r = 0$ is diffeomorphic to the hypersurface $r = \epsilon$ for some small $\epsilon > 0$, and the integral curves of $\xi_D$ hit the hypersurface $r = \epsilon$ at precisely one point. In other words, the hypersurface can be identified with the space of orbits of $\langle \xi_D \rangle$; that is, the hypersurface can be identified with the quotient of the homogeneous Lifshitz spacetime by $\langle \xi_D \rangle$. This quotient space is what we will call a $(d+1)$-dimensional Lifshitz–Weyl spacetime in this paper.

This story is analogous, albeit not precisely, to conformal geometry. One can think of an orientable conformal $n$-dimensional manifold as one whose frame bundle admits a reduction to the similitude group $CO(n) \cong SO(n) \times \mathbb{R}^+$. Here, we describe an anisotropic version of such a structure: namely, a $(d+1)$-dimensional manifold with a $CO(d)$-structure. This modification allows us to break the tangent bundle of such a manifold as a direct sum of a rank-$d$ vector bundle and a line bundle, and the dilatation subgroup of $CO(d)$ assigns a priori different weights to these bundles.

In this work we provide the first systematic study of Lifshitz Lie algebras, spacetimes and particles, as well as a first step in the classification of exotic aristotelian symmetries. We will now provide a summary of these results.

## 2 Summary and overview of the results

This work can roughly be separated into three parts:

1. a Lie algebraic part (Section 3) devoted to the classification of Lifshitz Lie algebras;

2. a geometric part (Sections 4 and 5) devoted to the classification of homogeneous spacetimes of Lifshitz Lie groups;

3. and a symplectic part (Section 6) which is devoted to the partial classification of homogeneous symplectic manifolds of Lifshitz Lie groups.

We will now provide a largely self-contained overview and summary of the main results of this paper and refer to the following sections for the proofs and further details.

## 2.1 Lie algebras

We start with an algebraic classification of **Lifshitz Lie algebras**, which by definition are Lie algebras which contain a rotational $\mathfrak{so}(d)$ subalgebra spanned by **J** under which the remaining generators transform as one vector **P** and two scalars H and D. This definition indeed leads to the prototypical Lifshitz symmetries but, as we will see, also to other interesting algebras and spacetimes which do not necessarily share the same interpretation. The definition fixes the Lie brackets

$$
\begin{aligned}
[J_{ab}, J_{cd}] &= \delta_{bc} J_{ad} - \delta_{ac} J_{bd} - \delta_{bd} J_{ac} + \delta_{ad} J_{bc}\,, \\
[J_{ab}, P_c] &= \delta_{bc} P_a - \delta_{ac} P_b\,, \\
[J_{ab}, H] &= 0\,, \\
[J_{ab}, D] &= 0\,.
\end{aligned}
\tag{5}
$$

The remaining undetermined Lie brackets are only subject to the Jacobi identity. Up to isomorphism there are seven classes of Lifshitz Lie algebras, which are listed in Table 1 for all $d \geqslant 2$. Two of the classes consist of two inequivalent Lie algebras distinguished by a sign and one of the classes is a one-parameter ($z$) family, as in equation (1). This Lie algebra, denoted $\mathfrak{a}_3^z$ here, is the prototypical Lifshitz algebra and it motivates our generalisations. The details of the classification are given in Section 3.1, 3.2, 3.3 and 3.4. Their interrelations are visualised in Figure 1 and interpreted geometrically in Section 4.3.

Table 1: Lifshitz Lie algebras in $d \geqslant 2$.

| Label | d | Nonzero Lie brackets in addition to $[J, J] = J$ and $[J, P] = P$ | | | Comments |
|---|---|---|---|---|---|
| $\mathfrak{a}_1$ | $\geqslant 2$ | | | | $\mathfrak{iso}(d) \oplus \mathbb{R}^2$ |
| $\mathfrak{a}_2$ | $\geqslant 2$ | $[D, H] = H$ | | | |
| $\mathfrak{a}_3^z$ | $\geqslant 2$ | $[D, H] = zH$ | $[D, P_a] = P_a$ | | $z \in \mathbb{R}$ |
| $\mathfrak{a}_4^\pm$ | $\geqslant 2$ | | | $[P_a, P_b] = \pm J_{ab}$ | $\mathfrak{so}(d, 1) \oplus \mathbb{R}^2,\ \mathfrak{so}(d+1) \oplus \mathbb{R}^2$ |
| $\mathfrak{a}_5^\pm$ | $\geqslant 2$ | $[D, H] = H$ | | $[P_a, P_b] = \pm J_{ab}$ | |
| $\mathfrak{a}_6$ | 2 | | | $[P_a, P_b] = \epsilon_{ab} H$ | |
| $\mathfrak{a}_7$ | 2 | $[D, H] = 2H$ | $[D, P_a] = P_a$ | $[P_a, P_b] = \epsilon_{ab} H$ | |

One might argue that it would be natural to restrict our attention to Lifshitz Lie algebras where none of the scalar generators, D or H, is central. Under such a restriction we would ignore, as the reader is free to do, the Lie algebras $\mathfrak{a}_1$, $\mathfrak{a}_3^{z=0}$, $\mathfrak{a}_4^\pm$ and $\mathfrak{a}_6$. We nevertheless keep them for completeness and uniformity, but we will see that they are mostly (rather trivial) generalisations of the aristotelian algebras and spacetimes without a particularly close connection to the prototypical Lifshitz algebra and spacetime.

A natural algebraic question, especially with our later applications in mind, is if the Lie algebras admit nontrivial central extensions. Some do, as we show in Section 3.5 and as summarised in Table 6.

## 2.2 Homogeneous spacetimes

In Section 4 we classify homogeneous spacetimes of the Lifshitz Lie algebras. These are smooth manifolds (with temporal and spatial directions) on which the Lifshitz Lie groups act transitively. Intuitively, the spacetimes look the same at every point, i.e., there are no preferred points. Well-known homogeneous spacetimes that therefore also have these properties are the lorentzian Minkowski and (anti) de Sitter spacetimes.

Homogeneous spaces of a Lie group G are described infinitesimally by Klein pairs $(\mathfrak{g}, \mathfrak{h})$ consisting the Lie algebra $\mathfrak{g}$ of G and a Lie subalgebra $\mathfrak{h}$ integrating to a closed subgroup of G.

Therefore a practical way to classify homogeneous spaces of G is to classify the Klein pairs $(\mathfrak{g}, \mathfrak{h})$. In practice this means searching for Lie admissible subalgebras $\mathfrak{h}$ of the Lie algebras in Table 1. It is only once we realise the Lie algebra as vector fields in a homogeneous space that we may assign a geometric interpretation to the generators in $\mathfrak{g}$, be it as rotations, translations in both space and time or (generalised) dilatations. We now discuss the three kinds of homogeneous spacetimes of Lifshitz Lie groups that we discuss in this paper.

### 2.2.1 $(d + 2)$-dimensional Lifshitz spacetimes

Here we quotient by the rotational part of the algebra, i.e., $\mathfrak{h}$ is spanned by $\mathbf{J}$. This leads to $(d + 2)$-dimensional spacetimes of which, loosely speaking and following the discussion in the Introduction, one direction could be interpreted as being holographic. These homogeneous Lifshitz spacetimes are summarised in Table 2 and there is one for each of the Lie algebras of Table 1. The details can be found in Section 4.1.

> Table 2: $(d+2)$-dimensional homogeneous Lifshitz spacetimes for $d \geqslant 2$. S is the static aristotelian spacetime, $\mathbb{E}^d$ is the d-dimensional euclidean space, TS the torsional static aristotelian spacetime, $\mathbb{H}^d$ is d-dimensional hyperbolic space, $\mathbb{S}^d$ is the d-dimensional sphere, $\mathbb{G}$ is the simply-connected two-dimensional Lie group whose Lie algebra is $[D, H] = H$ and $\mathbb{N}$ is the simply-connected three-dimensional Heisenberg group. See Table 5 for more details on the aristotelian geometries.

| L# | $\mathfrak{a}$ | d | Homogeneous space |
|----|----|----|----|
| 1 | $\mathfrak{a}_1$ | $\geqslant 2$ | $\mathsf{S} \times \mathbb{R} \cong \mathbb{E}^d \times \mathbb{R}^2$ |
| 2 | $\mathfrak{a}_2$ | $\geqslant 2$ | $\mathbb{E}^d \times \mathbb{G}$ |
| $3_{z=0}$ | $\mathfrak{a}_3^{z=0}$ | $\geqslant 2$ | $\mathsf{TS} \times \mathbb{R}$ |
| $3_{z\neq 0}$ | $\mathfrak{a}_3^{z\neq 0}$ | $\geqslant 2$ | Lifshitz spacetime |
| $4_\pm$ | $\mathfrak{a}_4^\pm$ | $\geqslant 2$ | $\mathbb{H}^d \times \mathbb{R}^2$, $\mathbb{S}^d \times \mathbb{R}^2$ |
| $5_\pm$ | $\mathfrak{a}_5^\pm$ | $\geqslant 2$ | $\mathbb{H}^d \times \mathbb{G}$, $\mathbb{S}^d \times \mathbb{G}$ |
| 6 | $\mathfrak{a}_6$ | 2 | $\mathbb{N} \times \mathbb{R}$ |
| 7 | $\mathfrak{a}_7$ | 2 | $\mathbb{N}$ fibration over $\mathbb{R}$ |

They all share the same invariants of low rank: namely, H and D, their dual one-forms $\eta, \delta$, as well as the degenerate metric $\pi^2$ and degenerate co-metric $P^2$. As we discuss in Section 5.1 one can use these invariants to construct nondegenerate metrics, in particular we explicitly recover the prototypical Lifshitz metric (2).

There is another class of homogeneous spaces that we can interpret as spacetimes of the Lifshitz algebras. When we quotient by one additional scalar generator we obtain $(d + 1)$-dimensional spacetimes and they are the subject of Section 4.2. Here we must discriminate between two cases depending on whether or not the Lifshitz Lie group acts effectively.[1] When the action is effective, we interpret all symmetries as spacetime symmetries and we are led to what we term Lifshitz–Weyl spacetimes.

On the other hand, if the action is not effective, then one of the scalars does not act on the spacetime. These spacetimes are aristotelian and we may interpret the additional generator as a scalar charge, such as, for example, an electric charge. If the Lie algebra does not split as a direct sum of an aristotelian algebra and the one-dimensional subalgebra spanned by the scalar charge, we call it an exotic spacetime symmetry.

---

[1]An action is effective if every non-identity element of the group moves some point.

### 2.2.2 $(d + 1)$-dimensional Lifshitz–Weyl spacetimes

For Lifshitz–Weyl spacetimes we quotient by one additional generator (D in our notation), to obtain a $(d + 1)$-dimensional homogeneous space. The name of these spacetimes originates from the interpretation of D as a (generalised) dilatation and we have summarised them in Table 3.

Table 3: Effective $(d + 1)$-dimensional homogeneous Lifshitz–Weyl spacetimes in $d \geqslant 2$. The bases are such that the stabiliser subalgebra $\mathfrak{h}$ is spanned by **J** and D. These spacetimes admit a canonical torsion-free invariant connection which is either flat or not (as denoted in the final column).

| LW# | $\mathfrak{a}$ | $d$ | Nonzero Lie brackets in addition to $[\mathbf{J}, \mathbf{J}] = \mathbf{J}$ and $[\mathbf{J}, \mathbf{P}] = \mathbf{P}$ | | | Geometry |
|---|---|---|---|---|---|---|
| 1 | $\mathfrak{a}_2$ | $\geqslant 2$ | $[D, H] = H$ | | | $z = \infty$, flat |
| $2_z$ | $\mathfrak{a}_3^z$ | $\geqslant 2$ | $[D, H] = zH$ | $[D, \mathbf{P}] = \mathbf{P}$ | | $z$, flat |
| $3_\pm$ | $\mathfrak{a}_5^\pm$ | $\geqslant 2$ | $[D, H] = H$ | | $[\mathbf{P}, \mathbf{P}] = \pm \mathbf{J}$ | $z = \infty$, not flat |
| 4 | $\mathfrak{a}_7$ | $2$ | $[D, H] = 2H$ | $[D, \mathbf{P}] = \mathbf{P}$ | $[\mathbf{P}, \mathbf{P}] = H$ | $z = 2$, not flat |

These homogeneous spaces can be roughly interpreted as "going to the holographic boundary" with respect to the $(d + 2)$ Lifshitz spacetimes based on the same Lie algebra. As described in Section 5.2 the invariants of Lifshitz–Weyl spacetimes are rotational invariant tensors on M which transform according to some weight. We have summarised these conformal weights in Table 7.

### 2.2.3 $(d + 1)$-dimensional aristotelian spacetimes with scalar charge

For the $(d + 1)$-dimensional aristotelian spacetimes we again quotient by rotations and one scalar charge. In this case this scalar acts trivially on the spacetime and is therefore not interpretable as a dilatation or any other spacetime symmetry, but as a scalar charge Q. Consequently, the underlying geometry is not Lifshitz but aristotelian. Aristotelian geometries permit rotational, spatio-temporal translational symmetries, but no boosts nor dilations (see [30] for a classification of aristotelian Lie algebras and spacetimes). We have summarised all of them in Table 4 where one can see that that they fall into two classes.

The first class consists of Lifshitz Lie algebras which are isomorphic to a direct sum $\langle \mathbf{J}, H, \mathbf{P} \rangle \oplus \mathbb{R}Q$ of an aristotelian Lie algebra and the one-dimensional Lie algebra spanned by the scalar charge Q. The scalar charge decouples from the spacetime algebra and we think of such symmetries as trivial.

The second class is more interesting. It consists of Lifshitz Lie algebra where Q is acted on nontrivially by the spacetime symmetries. Hence although Q still acts trivially on the spacetime, its nontrivial interaction with the spacetime symmetries promotes it to an "exotic spacetime symmetry" in an underlying aristotelian geometry. We call these spacetime-dependent charges and we say they are exotic because the "usual" (non-exotic) internal charges decouple from the spacetime symmetries. In some cases, e.g., $E_4$, the scalar charge is a central extension, but not in all, e.g., $E_1$. It is interesting to note that the only consistent way to get exotic symmetries in generic dimension is via a charge in the $[H, Q]$ commutator. We provide further context concerning exotic spacetime symmetries and their connection to multipole algebras and fractons in the concluding Section 7.

## 2.3 Lifshitz particles

Finally, in Section 6, we analyse (classical) **Lifshitz particles**, equivalently the **elementary systems** with Lifshitz symmetry. These are homogeneous symplectic manifolds of a Lifshitz

Table 4: $(d+1)$-dimensional aristotelian spacetimes with one scalar charge in $d \geqslant 2$. The bases are such that the stabiliser subalgebra $\mathfrak{h}$ is spanned by $\mathbf{J}$ and $Q$. In all cases the underlying spacetime geometry is aristotelian. As denoted in the first column some of them are exotic symmetries ($\mathsf{E}_1$ to $\mathsf{E}_5$) where the charge is influenced by the aristotelian symmetries. The remaining cases are such that the Lie algebra is a direct sum of an aristotelian algebra and the charge. $\mathsf{S}$ is the static aristotelian spacetime, $\mathsf{TS}$ the torsional static aristotelian spacetime, $\mathbb{H}^d$ is $d$-dimensional hyperbolic space, $\mathbb{S}^d$ is the $d$-dimensional sphere and $\mathbb{N}$ is the simply-connected three-dimensional Heisenberg group. See Table 5 for more details on the aristotelian geometries.

| E# | $\mathfrak{a}$ | d | Nonzero Lie brackets in addition to $[\mathbf{J},\mathbf{J}]=\mathbf{J}$ and $[\mathbf{J},\mathbf{P}]=\mathbf{P}$ | | | Geometry |
|---|---|---|---|---|---|---|
| | $\mathfrak{a}_1$ | $\geqslant 2$ | | | | S |
| 1 | $\mathfrak{a}_2$ | $\geqslant 2$ | $[H,Q]=Q$ | | | S |
| | $\mathfrak{a}_3^{z=0}$ | $\geqslant 2$ | | $[H,\mathbf{P}]=\mathbf{P}$ | | TS |
| $2_{z\neq0}$ | $\mathfrak{a}_3^{z\neq0}$ | $\geqslant 2$ | $[H,Q]=zQ$ | $[H,\mathbf{P}]=\mathbf{P}$ | | TS |
| | $\mathfrak{a}_4^{\pm}$ | $\geqslant 2$ | | | $[\mathbf{P},\mathbf{P}]=\pm\mathbf{J}$ | $\mathbb{H}^d\times\mathbb{R},\mathbb{S}^d\times\mathbb{R}$ |
| $3_{\pm}$ | $\mathfrak{a}_5^{\pm}$ | $\geqslant 2$ | $[H,Q]=Q$ | | $[\mathbf{P},\mathbf{P}]=\pm\mathbf{J}$ | $\mathbb{H}^d\times\mathbb{R},\mathbb{S}^d\times\mathbb{R}$ |
| | $\mathfrak{a}_6$ | 2 | | | $[\mathbf{P},\mathbf{P}]=H$ | $\mathbb{N}$ |
| 4 | $\mathfrak{a}_6$ | 2 | | | $[\mathbf{P},\mathbf{P}]=Q$ | S |
| 5 | $\mathfrak{a}_7$ | 2 | $[H,Q]=2Q$ | $[H,\mathbf{P}]=\mathbf{P}$ | $[\mathbf{P},\mathbf{P}]=Q$ | TS |

Lie group. They can be thought as the mechanistic description of elementary particles. The geometric quantisation of these symplectic manifolds give rise (as is well known for Poincaré and Galilei groups) to unitary irreducible representations of the group. As described in more detail in Section 6.1 a simply-connected homogeneous symplectic manifold of a Lie group $G$ is the universal cover of a coadjoint orbit of $G$ or possibly of a one-dimensional central extension. One is therefore led to study the central extensions of the Lie algebras, see Section 3.5, and the structure of the coadjoint orbits. It is important to emphasise that coadjoint orbits are an intrinsic property of the Lie group and only once a choice of homogeneous spacetime is made can they be interpreted as the space of motions of classical particles in a spacetime.

We construct the coadjoint actions for all cases and highlight the structure of the coadjoint orbits. We put special emphasis on the prototypical case $\mathfrak{a}_3^{z\neq0}$ and provide the necessary information to perform the classification, if so desired.

## 3 Lifshitz Lie algebras

In this section we classify Lifshitz Lie algebras, but first a definition.[2]

**Definition 1.** A *Lifshitz Lie algebra* is a $(d(d+1)/2+2)$-dimensional real Lie algebra $\mathfrak{g}$ with basis $J_{ab}=-J_{ba}, P_a, H, D$, for $a,b=1,\dots,d$, such that $J_{ab}$ span a Lie subalgebra $\mathfrak{r}\cong\mathfrak{so}(d)$, under which $P_a$ transforms as a vector and $H$ and $D$ transform as scalars.

---

[2]In [31], one of us introduced the notion of a "generalised Lifshitz algebra" to be a graded kinematical Lie algebra together with the grading element. The Lifshitz Lie algebras in the present paper will be seen to be graded aristotelian Lie algebras together with the grading element. This means that the "generalised Lifshitz algebras" in [31] are a special class of boost-extended Lifshitz Lie algebras, which will be the subject of a follow-up paper.

Unpacking this definition, we see that all such Lie algebras share the following Lie brackets:

$$
\begin{aligned}
[J_{ab}, J_{cd}] &= \delta_{bc} J_{ad} - \delta_{ac} J_{bd} - \delta_{bd} J_{ac} + \delta_{ad} J_{bc}\,, \\
[J_{ab}, P_c] &= \delta_{bc} P_a - \delta_{ac} P_b\,, \\
[J_{ab}, H] &= 0\,, \\
[J_{ab}, D] &= 0\,,
\end{aligned}
\tag{6}
$$

and the additional Lie brackets not involving $J_{ab}$ are subject only to the Jacobi identity.

If $d = 1$ then there are no rotations and hence any three-dimensional Lie algebra is Lifshitz. These were classified by Bianchi [32] and we will not discuss them further in this paper, except briefly when discussing the associated spacetimes. It will be convenient to break the discussion of the $d > 1$ algebras into three cases: $d = 2$, $d = 3$ and $d > 3$. For the following it will be useful to recall the aristotelian spacetimes.

The classification of aristotelian Lie algebras and aristotelian homogeneous spacetimes was done in [30], which we recall in Table 5. The notation is as in [30, Table 2], except that we have introduced $\mathbb{N}$ (as in "Nilmanifold") for the Heisenberg group which in [30] was called "A24". Since every aristotelian Lie algebra gives rise to a unique aristotelian Lie pair, this Table provides both the Lie algebras and the classification of simply-connected aristotelian spacetimes up to isomorphism.

Table 5: Aristotelian Lie algebras and simply-connected homogeneous $(d + 1)$-dimensional aristotelian spacetimes. The bases are such that the subalgebra $\mathfrak{h}$ of the spacetimes is spanned by $\mathbf{J}$.

| Name | $d$ | Nonzero Lie brackets in addition to $[\mathbf{J}, \mathbf{J}] = \mathbf{J}$ and $[\mathbf{J}, \mathbf{P}] = \mathbf{P}$ | Comments |
|---|---|---|---|
| S | $\geqslant 0$ | | static |
| TS | $\geqslant 1$ | $[H, \mathbf{P}] = \mathbf{P}$ | torsional static |
| $\mathbb{R} \times \mathbb{H}^d$ | $\geqslant 2$ | $[\mathbf{P}, \mathbf{P}] = \mathbf{J}$ | |
| $\mathbb{R} \times \mathbb{S}^d$ | $\geqslant 2$ | $[\mathbf{P}, \mathbf{P}] = -\mathbf{J}$ | |
| $\mathbb{N}$ | $2$ | $[\mathbf{P}, \mathbf{P}] = H$ | Heisenberg group |

## 3.1 Lifshitz algebras with $d > 3$

If $d > 3$, then the $\mathfrak{so}(d)$-equivariance of the brackets — equivalently, the Jacobi identity involving $J_{ab}$ — implies that the brackets involving $D$ are given by

$$
[D, H] = aH + bD \qquad \text{and} \qquad [D, P_a] = \lambda P_a\,.
\tag{7}
$$

In particular, $D$ and $H$ span a two-dimensional Lie subalgebra. Up to isomorphisms, there are precisely two such Lie algebras and we can therefore assume that $[D, H] = aH$, where $a \in \{0, 1\}$. In any case we see that $[D, -]$ is diagonal and hence $D$ is a grading element. Since $D$ does not appear in the RHS of any Lie brackets, we conclude that $D$ is a grading element for the aristotelian Lie subalgebra spanned by the remaining generators: $J_{ab}, P_a, H$.

## 3.2 Lifshitz algebras with $d = 3$

Now let $d = 3$. The $\mathfrak{so}(3)$-equivariance of the Lie brackets restricts their form to

$$
\begin{aligned}
[H, P_a] &= \alpha \epsilon_{abc} J^{bc} + \beta P_a\,, \\
[Z, P_a] &= \gamma \epsilon_{abc} J^{bc} + \delta P_a\,, \\
[D, H] &= aH\,, \\
[P_a, P_b] &= \varepsilon \epsilon_{abc} P^c + \eta J_{ab}\,.
\end{aligned}
\tag{8}
$$

The terms with coefficients $\alpha, \gamma, \varepsilon$ are unique to $d = 3$. We will show that they can be set to zero, which extends the results of $d > 3$ also to $d = 3$. We start with the observation that by shifting $P_a \mapsto P_a + \frac{\varepsilon}{4}\epsilon_{abc}J^{bc}$, we can (and will) set $\varepsilon = 0$ without loss of generality. Since we are in $d = 3$, it is more convenient to change basis to $J_{ab} = -\epsilon_{abc}J^c$, where $J_a = -\frac{1}{2}\epsilon_{abc}J^{bc}$. In this way, $[J_a, J_b] = \epsilon_{abc}J^c$, et cetera. Independently of the $[D, H]$ bracket, the $[H, \mathbf{P}, \mathbf{P}]$ Jacobi forces $\alpha = 0$ and the $[D, \mathbf{P}, \mathbf{P}]$ Jacobi forces $\gamma = 0$. Since $\alpha = \gamma = 0$, we see that the possible new terms in $d = 3$ do not arise and hence the results for $d > 3$ also apply to $d = 3$. In other words, $J_a, P_a, H$ span an aristotelian Lie algebra and $D$ is a grading element.

### 3.3 Lifshitz algebras with $d = 2$

We now consider the case $d = 2$. Now $J_{ab} = -\epsilon_{ab}J$ spans $\mathfrak{r} \cong \mathfrak{so}(2)$, which is abelian. The (non-zero) brackets are now

$$
\begin{aligned}
[J, P_a] &= \epsilon_{ab}P_b\,, \\
[P_a, P_b] &= \epsilon_{ab}(\alpha J + \beta H + \gamma D)\,, \\
[D, H] &= aJ + bH + cD\,, \\
[D, P_a] &= \delta P_a + \xi\epsilon_{ab}P_b\,, \\
[H, P_a] &= \theta P_a + \psi\epsilon_{ab}P_b\,.
\end{aligned}
\tag{9}
$$

The first observation is that we can set $\xi = \psi = 0$ without loss of generality by redefining $D \mapsto D - \xi J$ and $H \mapsto H - \psi J$. Having done that, there are three Jacobi identities which need to be checked: $[H, \mathbf{P}, \mathbf{P}]$, $[D, \mathbf{P}, \mathbf{P}]$ and $[D, H, \mathbf{P}]$. (The $[\mathbf{P}, \mathbf{P}, \mathbf{P}]$ Jacobi identity is identically satisfied because there are only two $P_a$.) Since $[D, -]$ and $[H, -]$ act diagonally on $P_a$, $[[D, H], -]$ must act trivially. This says that $J$ cannot appear in $[D, H]$ and hence $a = 0$. In addition, we see that $b\theta + c\delta = 0$. Since $a = 0$, $D, H$ span a two-dimensional Lie algebra and hence we can change basis so that $[D, H] = bH$ where $b \in \{0, 1\}$. Therefore again we see that $D$ is a grading element for the aristotelian Lie algebra spanned by $J, P_a, H$.

### 3.4 Summary

In summary, we have seen that every Lifshitz Lie algebra is obtained by adjoining a grading element to an aristotelian Lie algebra. It is a now simple matter to determine the possible gradings of the aristotelian Lie algebras in Table 5.

**Theorem 1.** *Every Lifshitz Lie algebra is isomorphic to precisely one of the Lie algebras in Table 1.*

*Proof.* As explained above, we need to determine the possible gradings of the aristotelian Lie algebras. These Lie algebras are in one-to-one correspondence with the homogeneous aristotelian spacetimes in Table 5. We go through them in turn, using as labels the names of the corresponding spacetimes.

(S) Any grading is possible: $[D, \mathbf{P}] = \lambda\mathbf{P}$ and $[D, H] = \mu H$. We distinguish three cases:

1. If $\lambda = \mu = 0$, we obtain $\mathfrak{a}_1$ in Table 1, which is simply the direct sum $\mathfrak{s} \oplus \mathbb{R}D$, where $\mathfrak{s}$ is the static aristotelian Lie algebra.

2. If $\lambda = 0$ but $\mu \neq 0$, we may rescale $D \mapsto \mu^{-1}D$ and we are left with $[D, \mathbf{P}] = 0$ and $[D, H] = H$, which could be thought of as the case $z = \infty$ of the Lifshitz Lie algebra (4). We label it $\mathfrak{a}_2$ in Table 1.

3. If $\lambda \neq 0$, we may rescale $D \mapsto \lambda^{-1}D$ so that $[D, \mathbf{P}] = \mathbf{P}$ and, letting $z := \mu/\lambda$, $[D, H] = zH$. This is the Lifshitz algebra in equation (4), which we label $\mathfrak{a}_3^z$ in Table 1.

(TS) In this case $[D, H] = 0$ and $[D, P] = \lambda P$. We must distinguish two cases, according to whether or not $\lambda = 0$.

    1. If $\lambda = 0$, we may rename $H \leftrightarrow D$ to arrive at a Lie algebra isomorphic to $\mathfrak{a}_3^{z=0}$.

    2. If $\lambda \neq 0$, we may rescale $D \mapsto \lambda^{-1}D$ and arrive at $[D, P] = P$ and then change basis $H \mapsto H - D$, so that $[H, P] = 0$. The resulting Lie algebra is also isomorphic to $\mathfrak{a}_3^{z=0}$.

$(\mathbb{R} \times \mathbb{H}^d)$ Here $[D, P] = 0$, so we have $[D, H] = \mu H$ and we must distinguish two cases depending on whether or not $\mu = 0$. As in the previous case, $\mu = 0$ says $D$ is central and gives a direct sum Lie algebra, whereas if $\mu \neq 0$, we may rescale $D$ so that $[D, H] = H$. The former is labelled $\mathfrak{a}_4^+$ in Table 1 and the latter is labelled $\mathfrak{a}_5^+$.

$(\mathbb{R} \times \mathbb{S}^d)$ This case is similar to the above and leads to the Lie algebras $\mathfrak{a}_4^-$ and $\mathfrak{a}_5^-$ in Table 1.

$(\mathbb{N})$ The only compatible grading is $[D, P] = \lambda P$ and $[D, H] = 2\lambda H$. If $\lambda = 0$ we have the Lie algebra $\mathfrak{a}_6$ in Table 1 and if $\lambda \neq 0$, we may rescale $D \mapsto \lambda^{-1}D$ so that $[D, P] = P$ and $[D, H] = 2H$, which is Lie algebra $\mathfrak{a}_7$.

$\qquad\qquad\qquad\qquad\qquad\qquad\qquad\qquad\qquad\qquad\qquad\qquad\qquad\qquad\qquad\quad$ $\square$

The Lifshitz Lie algebras are related via contractions which are depicted in Figure 1. Since the homogeneous Lifshitz spacetimes are in one-to-one correspondence with the Lifshitz Lie algebras, that figure also depicts limits between the spacetimes. Every node in the figure represents an isomorphism class of Lifshitz Lie algebras, with the red nodes only arising when $d = 2$. The thin edges correspond to contractions between the algebras, whereas the thick edge is the continuum $\mathfrak{a}_3^z$. We see that $\lim_{z \to \pm\infty} \mathfrak{a}_3^z = \mathfrak{a}_2$, so that $z$ should really be thought of parametrising a circle. We can make this manifest by defining $z = \cot\frac{\theta}{2}$ with $\theta \in [0, 2\pi]$. There are two distinguished points in this circle: $\theta = 2\cot^{-1} 2$, corresponding to $z = 2$, and $\theta = 0$ (equivalently, $\theta = 2\pi$) which corresponds to $z = \infty$.

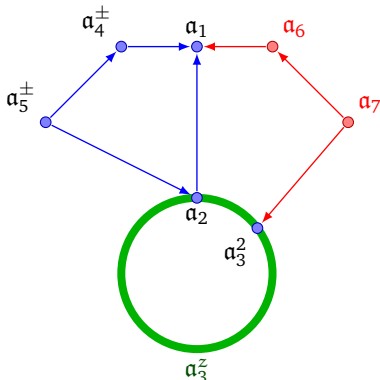

Figure 1: Contractions between Lifshitz Lie algebras.

(The red nodes and their corresponding arrows only exist if $d = 2$.)

### 3.5 Central extensions

As we will recall in Section 6, homogeneous symplectic manifolds (elementary particles in the language of Souriau [33]) of a Lie group $G$ are given locally by coadjoint orbits of $G$ or possibly of a one-dimensional central extension of $G$. As a first step in the determination of homogeneous symplectic manifolds of the Lifshitz Lie groups, we will work out the central extensions of the Lifshitz Lie algebras in Table 1.

Central extensions of a Lie algebra $\mathfrak{g}$ are classified by the second Chevalley–Eilenberg cohomology group $H^2(\mathfrak{g})$ (with values in the trivial one-dimensional representation). Since a Lifshitz Lie algebra $\mathfrak{g}$ contains a rotational subalgebra $\mathfrak{r}$ which acts reducibly on the Chevalley–Eilenberg complex and trivially on the cohomology, $H^2(\mathfrak{g})$ can be calculated from the (typically much smaller) subcomplex consisting of $\mathfrak{r}$-invariant cochains.

For the purposes of this paper, namely the determination of homogeneous symplectic manifolds of a Lifshitz Lie group there is one additional subtlety. Not every central extension of the Lie algebra integrates to a central extension of the Lie group G; although they always do if we were to take G to be simply connected. For example, it follows from $[J, D] = Z$, that the adjoint action of J on D integrates to

$$\mathrm{Ad}_{\exp(\theta J)} D = \exp(\theta\, \mathrm{ad}_J)D = D + \theta Z \,, \tag{10}$$

which is not periodic in $\theta$. Hence it is not a central extension of the Lie group G where the rotational group is SO(2); although it is a central extension of the universal covering group where the "rotation" subgroup is $\mathbb{R}$. As explained briefly in Section 6, we restrict[3] to compact rotations and therefore these central extension are not relevant for the determination of homogeneous symplectic manifolds with symmetry group G. This means we can actually restrict ourselves not just to the $\mathfrak{r}$-invariant subcomplex, but even further to the $\mathfrak{r}$-basic subcomplex which calculates the relative cohomology $H^2(\mathfrak{g}; \mathfrak{r})$. We choose to calculate $H^2(\mathfrak{g})$ below, since this may be of independent interest, but later in the paper we shall only be interested in some of these central extensions.

By introducing a suitable parameter, we may treat the Lie algebras $\mathfrak{a}_1$, $\mathfrak{a}_2$ and $\mathfrak{a}_3^z$ together. Similarly we may treat $\mathfrak{a}_4^\pm$ and $\mathfrak{a}_5^\pm$ together and also $\mathfrak{a}_6$ and $\mathfrak{a}_7$ together. The results are summarised in Table 6, where central extensions in $H^2(\mathfrak{g})$ which are not in $H^2(\mathfrak{g}; \mathfrak{r})$ have been parenthesised and we shall ignore them in the remainder of the paper.

Table 6: Nontrivial central extensions of the Lifshitz Lie algebras in Table 1. This table shows the central extensions of the Lifshitz algebras $\mathfrak{g}$. The dimension of the cohomology group $H^2(\mathfrak{g})$ depends in some cases on the dimension d of the Lie algebra $\mathfrak{g}$. For compact rotations, the central extensions in parentheses do not integrate to the group and are related to the relative cohomology $H^2(\mathfrak{g}; \mathfrak{r})$, where $\mathfrak{r}$ is spanned by the rotations.

| $\mathfrak{g}$ | d | dim $H^2(\mathfrak{g})$ | dim $H^2(\mathfrak{g}, \mathfrak{r})$ | Nonzero Lie brackets of central extensions | | | |
|---|---|---|---|---|---|---|---|
| $\mathfrak{a}_1$ | $\geqslant 3$ | 1 | 1 | $[D, H] = Z$ | | | |
| | 2 | 4 | 2 | $[D, H] = Z$ | $[P_a, P_b] = \epsilon_{ab} Z_P$ | $([J, D] = Z_D)$ | $([J, H] = Z_H)$ |
| $\mathfrak{a}_2$ | $\geqslant 3$ | 0 | 0 | | | | |
| | 2 | 2 | 1 | | $[P_a, P_b] = \epsilon_{ab} Z_P$ | $([J, D] = Z_D)$ | |
| $\mathfrak{a}_3^{z\neq 0}$ | $\geqslant 3$ | 0 | 0 | | | | |
| | 2 | 1 | 0 | | | $([J, D] = Z_D)$ | |
| $\mathfrak{a}_3^{z=0}$ | $\geqslant 3$ | 1 | 1 | $[D, H] = Z$ | | | |
| | 2 | 3 | 1 | $[D, H] = Z$ | | $([J, D] = Z_D)$ | $([J, H] = Z_H)$ |
| $\mathfrak{a}_4^\pm$ | $\geqslant 2$ | 1 | 1 | $[D, H] = Z$ | | | |
| $\mathfrak{a}_5^\pm$ | $\geqslant 2$ | 0 | 0 | | | | |
| $\mathfrak{a}_6$ | 2 | 1 | 0 | | | $([J, D] = Z_D)$ | |
| $\mathfrak{a}_7$ | 2 | 1 | 0 | | | $([J, D] = Z_D)$ | |

---

[3]Let us however note that we could envision interesting physics for the case where one drops this restriction: for example, Lifshitz anyons in $d = 2$ as in [34].

### 3.5.1 Central extensions of $\mathfrak{a}_1$, $\mathfrak{a}_2$ and $\mathfrak{a}_3^z$

We introduce a parameter $w \in \{0, 1\}$ and modify the Lie brackets by $[D, P] = wP$. We can then treat all three Lie algebras simultaneously with $\mathfrak{a}_1$ corresponding to $w = z = 0$, $\mathfrak{a}_2$ corresponding to $w = 0, z = 1$ and $\mathfrak{a}_3^z$ corresponding to $w = 1$. Any other choice of $w, z$ is related to one of these by rescaling the generator D.

Let $\mathfrak{g}$ (depending on $w, z$) be one of these Lie algebras. A basis for $\mathfrak{g}$ is $J_{ab}, P_a, H, D$ and the canonical dual basis for $\mathfrak{g}^*$ is $\lambda^{ab}, \pi^a, \eta, \delta$. We will use the notation $\langle \cdots \rangle$ to mean the real span of the enclosed vectors, so that $\mathfrak{g} = \langle J_{ab}, P_a, H, D \rangle$ and $\mathfrak{g}^* = \langle \lambda^{ab}, \pi^a, \eta, \delta \rangle$. We let $\mathfrak{r} = \langle J_{ab} \rangle$. We are interested in the following fragment of the $\mathfrak{r}$-invariant subcomplex of the Chevalley–Eilenberg complex

$$(\mathfrak{g}^*)^{\mathfrak{r}} \xrightarrow{\ \partial\ } (\wedge^2 \mathfrak{g}^*)^{\mathfrak{r}} \xrightarrow{\ \partial\ } (\wedge^3 \mathfrak{g}^*)^{\mathfrak{r}}, \tag{11}$$

where

$$(\mathfrak{g}^*)^{\mathfrak{r}} = \left\langle \eta, \delta, \tfrac{1}{2}\epsilon_{ab}\lambda^{ab} \right\rangle, \tag{12}$$

and

$$\begin{aligned}
(\wedge^2 \mathfrak{g}^*)^{\mathfrak{r}} = \Big\langle &\eta \wedge \delta, \tfrac{1}{2}\epsilon_{ab}\lambda^{ab} \wedge \delta, \tfrac{1}{2}\epsilon_{ab}\lambda^{ab} \wedge \eta, \\
&\tfrac{1}{2}\epsilon_{ab}\pi^a \wedge \pi^b, \tfrac{1}{2}\epsilon_{abc}\lambda^{ab} \wedge \pi^c, \tfrac{1}{8}\epsilon_{abcd}\lambda^{ab} \wedge \lambda^{cd} \Big\rangle,
\end{aligned} \tag{13}$$

and where a term involving the Levi-Civita $\epsilon$ symbol only appears in the relevant dimension.

The Chevalley–Eilenberg differential on generators is given by

$$\partial\lambda^{ab} = -\lambda^a{}_c \wedge \lambda^{cb}, \quad \partial\pi^a = -\lambda^a{}_b \wedge \pi^b - w\delta \wedge \pi^a, \quad \delta\eta = -z\delta \wedge \eta \quad \text{and} \quad \partial\delta = 0. \tag{14}$$

We see that the space of 2-coboundaries $B^2(\mathfrak{g})^{\mathfrak{r}} = \partial(\mathfrak{g}^*)^{\mathfrak{r}} \subset (\wedge^2 \mathfrak{g}^*)^{\mathfrak{r}}$ is given by

$$B^2(\mathfrak{g})^{\mathfrak{r}} = \begin{cases} \langle \delta \wedge \eta \rangle, & \text{if } z \neq 0, \\ 0, & \text{if } z = 0. \end{cases} \tag{15}$$

We calculate $\partial : (\wedge^2 \mathfrak{g}^*)^{\mathfrak{r}} \to (\wedge^3 \mathfrak{g}^*)^{\mathfrak{r}}$ to obtain

$$\begin{aligned}
\partial(\delta \wedge \eta) &= 0, \\
\partial(\tfrac{1}{2}\epsilon_{ab}\lambda^{ab} \wedge \delta) &= 0, \\
\partial(\tfrac{1}{2}\epsilon_{ab}\lambda^{ab} \wedge \eta) &= z\epsilon_{ab}\lambda^{ab} \wedge \delta \wedge \eta, \\
\partial(\tfrac{1}{2}\epsilon_{ab}\pi^a \wedge \pi^b) &= -2w\epsilon_{ab}\pi^a \wedge \pi^b \wedge \delta, \\
\partial(\tfrac{1}{2}\epsilon_{abc}\lambda^{ab} \wedge \pi^c) &= -2\epsilon_{abc}\lambda^{ab} \wedge \pi^c \wedge \delta, \\
\partial(\tfrac{1}{8}\epsilon_{abcd}\lambda^{ab} \wedge \lambda^{cd}) &= -2\epsilon_{abcd}\lambda^a{}_e \wedge \lambda^{eb} \wedge \lambda^{cd}.
\end{aligned} \tag{16}$$

We see that the space of 2-cocycle $Z^2(\mathfrak{g})^{\mathfrak{r}} = \ker \partial : (\wedge^2 \mathfrak{g}^*)^{\mathfrak{r}} \to (\wedge^3 \mathfrak{g}^*)^{\mathfrak{r}}$ is given by

$$Z^2(\mathfrak{g})^{\mathfrak{r}} = \begin{cases} \left\langle \delta \wedge \eta, \tfrac{1}{2}\epsilon_{ab}\pi^a \wedge \pi^b, \tfrac{1}{2}\epsilon_{ab}\lambda^{ab} \wedge \delta, \tfrac{1}{2}\epsilon_{ab}\lambda^{ab} \wedge \eta \right\rangle, & \text{if } w = z = 0, \\ \left\langle \delta \wedge \eta, \tfrac{1}{2}\epsilon_{ab}\pi^a \wedge \pi^b, \tfrac{1}{2}\epsilon_{ab}\lambda^{ab} \wedge \delta \right\rangle, & \text{if } w = 0 \text{ and } z \neq 0, \\ \left\langle \delta \wedge \eta, \tfrac{1}{2}\epsilon_{ab}\lambda^{ab} \wedge \eta, \tfrac{1}{2}\epsilon_{ab}\lambda^{ab} \wedge \delta \right\rangle, & \text{if } w = 1 \text{ and } z = 0, \\ \left\langle \delta \wedge \eta, \tfrac{1}{2}\epsilon_{ab}\lambda^{ab} \wedge \delta \right\rangle, & \text{if } w = 1 \text{ and } z \neq 0. \end{cases} \tag{17}$$

Therefore we see that $H^2(\mathfrak{g}) = Z^2(\mathfrak{g})^{\mathfrak{r}}/B^2(\mathfrak{g})^{\mathfrak{r}}$ behaves quite differently in $d = 2$ and $d \geqslant 3$. The latter is given by

$$H^2(\mathfrak{g})_{d \geqslant 3} = \begin{cases} \langle [\delta \wedge \eta] \rangle, & \text{if } z = 0, \\ 0, & \text{if } z \neq 0, \end{cases} \tag{18}$$

where here and in the sequel square brackets denotes the cohomology class of the enclosed cocycle. We see that for $d \geqslant 3$, $\mathfrak{a}_2$ and $\mathfrak{a}_3^{z \neq 0}$ do not admit any nontrivial central extensions, whereas $\mathfrak{a}_1$ and $\mathfrak{a}_3^{z=0}$ admit a one-dimensional nontrivial central extension with bracket $[D, H] = Z$.

If $d = 2$ things are more complicated:

$$H^2(\mathfrak{g})_{d=2} = \begin{cases} \langle [\frac{1}{2}\epsilon_{ab}\lambda^{ab} \wedge \delta] \rangle, & \text{if } w = 1 \text{ and } z \neq 0, \\ \langle [\frac{1}{2}\epsilon_{ab}\lambda^{ab} \wedge \delta], [\frac{1}{2}\epsilon_{ab}\pi^a \wedge \pi^b] \rangle, & \text{if } w = 0 \text{ and } z \neq 0, \\ \langle [\delta \wedge \eta], [\frac{1}{2}\epsilon_{ab}\lambda^{ab} \wedge \delta], [\frac{1}{2}\epsilon_{ab}\lambda^{ab} \wedge \eta] \rangle, & \text{if } w = 1 \text{ and } z = 0, \\ \langle [\delta \wedge \eta], [\frac{1}{2}\epsilon_{ab}\lambda^{ab} \wedge \delta], [\frac{1}{2}\epsilon_{ab}\lambda^{ab} \wedge \eta], [\frac{1}{2}\epsilon_{ab}\pi^a \wedge \pi^b] \rangle, & \text{if } w = z = 0. \end{cases} \quad (19)$$

So that if $d = 2$, the Lifshitz Lie algebra $\mathfrak{a}_3^{z \neq 0}$ admits a nontrivial central extension with additional bracket $[J, D] = Z_D$; the Lie algebra $\mathfrak{a}_2$ admits a two-dimensional space of nontrivial central extensions with brackets $[J, D] = Z_D$ and $[P_a, P_b] = \epsilon_{ab}Z_P$; the Lie algebra $\mathfrak{a}_3^{z=0}$ admits a three-dimensional space of nontrivial central extensions with brackets $[D, H] = Z$, $[J, D] = Z_D$ and $[J, H] = Z_H$; and $\mathfrak{a}_1$ admits a four-dimensional space of nontrivial central extensions with brackets $[D, H] = Z$, $[J, D] = Z_D$, $[J, H] = Z_H$ and $[P_a, P_b] = \epsilon_{ab}Z_P$.

### 3.5.2 Central extensions of $\mathfrak{a}_4^{\pm}$ and $\mathfrak{a}_5^{\pm}$

We treat these two Lie algebras simultaneously by introducing a parameter $z \in \{0, 1\}$ and defining the bracket $[D, H] = zH$. If $z = 0$ we are in $\mathfrak{a}_4^{\pm}$ and if $z = 1$ we are in $\mathfrak{a}_5^{\pm}$. The bases for $\mathfrak{g}$ and $\mathfrak{g}^*$ are as in the previous section and the spaces of $\mathfrak{r}$-invariant 1- and 2-cochains are as before and given in equations (12) and (13), respectively. The Chevalley–Eilenberg differential is of course different and given on generators by

$$\partial\lambda^{ab} = -\lambda^a{}_c \wedge \lambda^{cb} \mp \pi^a \wedge \pi^b, \quad \partial\pi^a = -\lambda^a{}_b \wedge \pi^b, \quad \partial\eta = -z\delta \wedge \eta \quad \text{and} \quad \partial\delta = 0. \quad (20)$$

So that the space of 2-coboundaries is now given by

$$B^2(\mathfrak{g})^{\mathfrak{r}} = \langle z\delta \wedge \eta, \tfrac{1}{2}\epsilon_{ab}\pi^a \wedge \pi^b \rangle. \quad (21)$$

We calculate the differential on the 2-cochains and obtain

$$\begin{aligned} \partial(\delta \wedge \eta) &= 0, \\ \partial(\tfrac{1}{2}\epsilon_{ab}\lambda^{ab} \wedge \delta) &= \mp\tfrac{1}{2}\epsilon_{ab}\pi^a \wedge \pi^b \wedge \delta, \\ \partial(\tfrac{1}{2}\epsilon_{ab}\lambda^{ab} \wedge \eta) &= \mp\tfrac{1}{2}\epsilon_{ab}\pi^a \wedge \pi^b \wedge \eta + \tfrac{1}{2}z\epsilon_{ab}\lambda^{ab} \wedge \delta \wedge \eta, \\ \partial(\tfrac{1}{2}\epsilon_{ab}\pi^a \wedge \pi^b) &= 0, \\ \partial(\tfrac{1}{2}\epsilon_{abc}\lambda^{ab} \wedge \pi^c &= \mp\tfrac{1}{2}\epsilon_{abc}\pi^a \wedge \pi^b \wedge \pi^c, \\ \partial(\tfrac{1}{8}\epsilon_{abcd}\lambda^{ab} \wedge \lambda^{cd}) &= -\tfrac{1}{4}\epsilon_{abcd}\lambda^a{}_e \wedge \wedge\lambda^{eb} \wedge \lambda^{cd} \mp \tfrac{1}{4}\epsilon_{abcd}\pi^a \wedge \pi^b \wedge \lambda^{cd}, \end{aligned} \quad (22)$$

so that the space of 2-cocycles is

$$Z^2(\mathfrak{g})^{\mathfrak{r}} = \langle \eta \wedge \delta, \tfrac{1}{2}\epsilon_{ab}\pi^a \wedge \pi^b \rangle. \quad (23)$$

In summary, the cohomology is then

$$H^2(\mathfrak{g}) = \begin{cases} \langle [\eta \wedge \delta] \rangle, & \text{if } z = 0, \\ 0, & \text{if } z \neq 0. \end{cases} \quad (24)$$

Hence $\mathfrak{a}_4^{\pm}$ admits a one-dimensional non-trivial central extension with bracket $[D, H] = Z$ and $\mathfrak{a}_5^{\pm}$ admits none.

### 3.5.3 Central extensions of $\mathfrak{a}_6$ and $\mathfrak{a}_7$

Finally, we introduce a parameter $w \in \{0, 1\}$ in the brackets $[D, P] = wP$ and $[D, H] = 2wH$ so that if $w = 0$ we are in $\mathfrak{a}_6$ and if $w = 1$ we are in $\mathfrak{a}_7$. We are in $d = 2$ here, so that the $\mathfrak{r}$-invariant 1- and 2-cochains are

$$(\mathfrak{g}^*)^{\mathfrak{r}} = \left\langle \eta, \delta, \tfrac{1}{2}\epsilon_{ab}\lambda^{ab} \right\rangle , \tag{25}$$

and

$$(\wedge^2 \mathfrak{g}^*)^{\mathfrak{r}} = \left\langle \eta \wedge \delta, \tfrac{1}{2}\epsilon_{ab}\lambda^{ab} \wedge \delta, \tfrac{1}{2}\epsilon_{ab}\lambda^{ab} \wedge \eta, \tfrac{1}{2}\epsilon_{ab}\pi^a \wedge \pi^b \right\rangle . \tag{26}$$

The action of the differential on generators is now

$$\partial \lambda^{ab} = 0, \quad \partial \eta = -2w\delta \wedge \eta - \tfrac{1}{2}\epsilon_{ab}\pi^a \wedge \pi^b ,$$
$$s\partial \pi^a = -\lambda^a{}_b \wedge \pi^b - w\delta \wedge \pi^a \quad \text{and} \quad \partial \delta = 0 . \tag{27}$$

The 2-coboundaries are then

$$B^2(\mathfrak{g})^{\mathfrak{r}} = \left\langle \tfrac{1}{2}\epsilon_{ab}\pi^a \wedge \pi^b + 2w\delta \wedge \eta \right\rangle . \tag{28}$$

We calculate the differential on $\mathfrak{r}$-invariant cochains to be

$$\partial(\eta \wedge \delta) = -\tfrac{1}{2}\epsilon_{ab}\pi^a \wedge \pi^b \wedge \delta ,$$
$$\partial(\tfrac{1}{2}\epsilon_{ab}\lambda^{ab} \wedge \eta) = w\epsilon_{ab}\lambda^{ab} \wedge \delta \wedge \eta + \tfrac{1}{2}\lambda^{ab} \wedge \pi_a \wedge \pi_b ,$$
$$\partial(\tfrac{1}{2}\epsilon_{ab}\lambda^{ab} \wedge \delta) = 0 ,$$
$$\partial(\tfrac{1}{2}\epsilon_{ab}\pi^a \wedge \pi^b) = -w\epsilon_{ab}\pi^a \wedge \pi^b \wedge \delta . \tag{29}$$

We see that the space of 2-cocycles is given by

$$Z^2(\mathfrak{g})^{\mathfrak{r}} = \left\langle \tfrac{1}{2}\epsilon_{ab}\pi^a \wedge \pi^b + 2w\delta \wedge \eta, \tfrac{1}{2}\epsilon_{ab}\lambda^{ab} \wedge \delta \right\rangle , \tag{30}$$

so that the cohomology is given by

$$H^2(\mathfrak{g}) = \left\langle [\tfrac{1}{2}\epsilon_{ab}\lambda^{ab} \wedge \delta] \right\rangle . \tag{31}$$

Therefore both $\mathfrak{a}_6$ and $\mathfrak{a}_7$ admit a one-dimensional nontrivial central extension with bracket $[J, D] = Z_D$.

## 4 Spatially isotropic homogeneous Lifshitz spacetimes

In this section we classify the homogeneous spacetimes associated to the Lifshitz algebras in Table 1. As discussed in the introduction, we are interested in two kinds of homogeneous spacetimes. Firstly, we have the $(d + 2)$-dimensional Lifshitz spacetimes which are described infinitesimally by Klein pairs of the form $(\mathfrak{a}, \mathfrak{r})$, where $\mathfrak{a}$ is a Lifshitz algebra and $\mathfrak{r} \cong \mathfrak{so}(d)$ is the rotational subalgebra, which are the subject of Section 4.1. Secondly, we have the $(d + 1)$-dimensional Lifshitz–Weyl spacetimes whose Klein pairs are now $(\mathfrak{a}, \mathfrak{h})$, where $\mathfrak{a}$ is again a Lifshitz algebra, but now $\mathfrak{h} \cong \mathfrak{co}(d) = \mathfrak{so}(d) \oplus \mathbb{R}$ is spanned by the rotations and one one of the scalars in the span of $D, H$. This is the subject of Section 4.2.

## 4.1 Homogeneous Lifshitz spacetimes

We now list the possible Klein pairs $(\mathfrak{a}, \mathfrak{r})$ where $\mathfrak{a}$ is one of the Lie algebras of Table 1 and $\mathfrak{r} \cong \mathfrak{so}(d)$ is the subalgebra spanned by $J_{ab}$. Clearly these are indexed by the Lie algebras $\mathfrak{a}$ in Table 1 themselves. We may easily identify the corresponding homogeneous spaces, partially from the classification in [30, Appendix A]. The results are tabulated in Table 2. The notation is such that $\mathbb{H}^d$ is d-dimensional hyperbolic space, $\mathbb{S}^d$ is the d-dimensional sphere, $\mathbb{E}^d$ is the d-dimensional euclidean space, S is the static aristotelian spacetime, TS the torsional static aristotelian spacetime, $\mathbb{N}$ is the simply-connected three-dimensional Heisenberg group (a three-dimensional aristotelian spacetime prosaically labelled A24 in [30]) and $\mathbb{G}$ is the simply-connected two-dimensional Lie group whose Lie algebra is $[D, H] = H$. For $d = 1$, since $\mathfrak{r} = 0$, the Klein pairs are simply the Lie algebras themselves, which are the Bianchi Lie algebras. The spacetimes are the simply-connected three-dimensional Lie groups, as in Bianchi's original paper [32].

The simply-connected four-dimensional homogeneous Lifshitz spacetime with Klein pair $(\mathfrak{a}_7, \mathfrak{r})$ may be identified with the simply-connected Lie group $\mathcal{G}$ whose Lie algebra $\mathfrak{g}$ is an extension-by-derivation of the three-dimensional Heisenberg Lie algebra $\mathfrak{n}$, so fitting into an exact sequence

$$0 \longrightarrow \mathfrak{n} \longrightarrow \mathfrak{g} \longrightarrow \mathbb{R}D \longrightarrow 0 \,. \tag{32}$$

The group $\mathcal{G}$ is foliated by copies of the Heisenberg group $\mathbb{N}$ and fibres over the real line.

## 4.2 Homogeneous Lifshitz–Weyl and aristotelian spacetimes with scalar charge

We now list the possible Klein pairs $(\mathfrak{a}, \mathfrak{h})$ where $\mathfrak{a}$ is one of the Lie algebras of Table 1 and $\mathfrak{h} = \mathfrak{r} \oplus \mathbb{R}S$ is the subalgebra spanned by $J_{ab}$ and a scalar S in the span of D and H. These Klein pairs describe homogeneous Lifshitz–Weyl spacetimes or aristotelian spacetimes with one scalar charge. For each Lie algebra $\mathfrak{a}$ in Table 1, we determine the possible one-dimensional scalar lines (spanned by S) up to the action of J-preserving automorphisms of $\mathfrak{a}$; that is, automorphisms of $\mathfrak{a}$ which are the identity on $\mathfrak{r}$. For $d = 1$, the Klein pairs were already classified (from the point of view of kinematical spacetimes) in [30, Section 3.4] and further studied in [35].

### 4.2.1 Klein pairs associated to $\mathfrak{a}_1$

The J-preserving automorphisms are given by

$$\mathbf{P} \mapsto \mu \mathbf{P} \,, \qquad H \mapsto aH + bD \qquad \text{and} \qquad D \mapsto cH + dD \,, \tag{33}$$

where $\mu \neq 0$ and $\begin{pmatrix} a & b \\ c & d \end{pmatrix}$ is invertible. Clearly, we can take any S to D via an automorphism. The resulting Klein pair is not effective, since D spans an ideal both of $\mathfrak{a}$ and of $\mathfrak{h}$. Quotienting by this ideal, we obtain a Klein pair $(\mathfrak{s}, \mathfrak{r})$ where $\mathfrak{s}$ is the aristotelian static Lie algebra and $\mathfrak{r}$ is the rotational subalgebra. As shown in [30], this is the Klein pair of the static aristotelian spacetime S.

### 4.2.2 Klein pairs associated to $\mathfrak{a}_2$

The J-preserving automorphisms are given by

$$\mathbf{P} \mapsto \mu \mathbf{P} \,, \qquad H \mapsto aH \qquad \text{and} \qquad D \mapsto D + cH \,, \tag{34}$$

where $\mu, a \neq 0$ and $c \in \mathbb{R}$. Suppose that $S = \alpha H + \beta D$. Then under such an automorphism, $S \mapsto (a\alpha + \beta c)H + \beta D$. If $\beta \neq 0$, then we can choose c so that $S = \beta D$ and if $\beta = 0$, then

$S = \alpha H$. In other words, we can take $\mathfrak{h} = \mathfrak{r} \oplus \mathbb{R}D$ or $\mathfrak{h} = \mathfrak{r} \oplus \mathbb{R}H$. The latter Klein pair is not effective, since H spans an ideal of both $\mathfrak{a}$ and $\mathfrak{h}$. Quotienting by this ideal gives the Klein pair of the aristotelian static spacetime S. The former Klein pair is effective and describes a homogeneous flat Lifshitz spacetime with $z = \infty$.

### 4.2.3 Klein pairs associated to $\mathfrak{a}_3^z$

Here the J-preserving automorphisms are given by

$$\mathbf{P} \mapsto \mu \mathbf{P}, \qquad H \mapsto \alpha H \qquad \text{and} \qquad D \mapsto D + cH, \tag{35}$$

where $\mu, \alpha \neq 0$ and $c \in \mathbb{R}$, and as before we have two choices of scalar lines up to automorphisms: $\mathfrak{h} = \mathfrak{r} \oplus \mathbb{R}D$ or $\mathfrak{h} = \mathfrak{r} \oplus \mathbb{R}H$. The latter Klein pair is not effective, since H spans an ideal of both $\mathfrak{a}$ and $\mathfrak{h}$. Quotienting by this ideal gives the Klein pair of the aristotelian torsional static spacetime TS. The former Klein pair is effective and describes a homogeneous flat Lifshitz–Weyl spacetime with scaling exponent $z \neq 0$.

### 4.2.4 Klein pairs associated to $\mathfrak{a}_4^\pm$

The J-preserving automorphisms are given by

$$\mathbf{P} \mapsto \pm \mathbf{P}, \qquad H \mapsto \alpha H + bD \qquad \text{and} \qquad D \mapsto cH + dD, \tag{36}$$

where $\begin{pmatrix} \alpha & b \\ c & d \end{pmatrix}$ is invertible. We may take $S = D$ without loss of generality, resulting in a non-effective Klein pair describing $\mathbb{H}^d \times \mathbb{R}$ or $\mathbb{S}^d \times \mathbb{R}$.

### 4.2.5 Klein pairs associated to $\mathfrak{a}_5^\pm$

The J-preserving automorphisms are given by

$$\mathbf{P} \mapsto \pm \mathbf{P}, \qquad H \mapsto \alpha H \qquad \text{and} \qquad D \mapsto D + cH, \tag{37}$$

where $\alpha \neq 0$ and $c \in \mathbb{R}$. There are two possibilities, namely $S = H$ and $S = D$. If $S = H$, the resulting Klein pair is non-effective and quotienting by the ideal generated by H gives either $\mathbb{H}^d \times \mathbb{R}$ or $\mathbb{S}^d \times \mathbb{R}$. If we take $S = D$ then we get an effective Klein pair corresponding to a generalised Lifshitz spacetime with $z = \infty$, but this time with curvature.

### 4.2.6 Klein pairs associated to $\mathfrak{a}_6$

The J-preserving automorphisms are given by

$$\mathbf{P} \mapsto \lambda \mathbf{P}, \qquad H \mapsto \lambda^2 H \qquad \text{and} \qquad D \mapsto \alpha D + bH, \tag{38}$$

with $\alpha, \lambda$ nonzero. As before, there are two possibilities: $S = D$ and $S = H$. Neither case is effective, quotienting by the ideal generated by D we obtain the Heisenberg group $\mathbb{N}$ as an aristotelian spacetime, whereas quotienting by the ideal generated by H we obtain the static aristotelian spacetime S.

### 4.2.7 Klein pairs associated to $\mathfrak{a}_7$

The J-preserving automorphisms are given by

$$\mathbf{P} \mapsto \lambda \mathbf{P}, \qquad H \mapsto cH \qquad \text{and} \qquad D \mapsto \alpha D + bH, \tag{39}$$

where $\alpha, c, \lambda$ are nonzero. As in the previous case, there are two possibilities: $S = H$ and $S = D$. If we take $S = H$, the resulting Klein pair is not non-effective and quotienting by the ideal generated by H recovers the torsional static aristotelian spacetime TS. Taking $S = D$ we get an effective Klein pair describing a Lifshitz–Weyl geometry with $z = 2$ and nonzero curvature.

#### 4.2.8 Summary

We summarise this discussion in Tables 3 and 4, which list the homogeneous Lifshitz–Weyl and aristotelian spacetimes with scalar charge, respectively. We list the Klein pairs $(\mathfrak{a}, \mathfrak{h})$ and we have changed basis so that the scalar S in $\mathfrak{h}$ is denoted D in the case it acts effectively or Q in case it does not.

The Lifshitz–Weyl spacetimes are all both reductive and symmetric. Being symmetric homogeneous spaces, they have a canonical torsion-free invariant connection, which is flat in the first two cases (LW1 and LW2$_z$) and not flat in the next two (LW3$_\pm$ and LW4).

### 4.3 Geometric interpretation of the limits

Having understood the nature of the homogeneous Lifshitz spacetimes, we may now give a geometric interpretation of some of the Lie algebra contractions in Figure 1. The contractions $\mathfrak{a}_5^\pm \to \mathfrak{a}_2$ are flat limits in that the round metric on $\mathbb{S}^d$ and the hyperbolic metric on $\mathbb{H}^d$ flatten to become the euclidean metric on $\mathbb{E}^d$. This interpretation also holds for the $(d+1)$-dimensional homogeneous aristotelian spacetimes with scalar charge. For the $(d+1)$-dimensional Lifshitz–Weyl spacetimes, the interpretation is slightly different. In this case the flat limit refers to the flatness of the canonical torsion-free invariant connection.

The contractions $\mathfrak{a}_5^\pm \to \mathfrak{a}_4^\pm$ and $\mathfrak{a}_2 \to \mathfrak{a}_1$ are such that the nonabelian Lie group $\mathbb{G}$ becomes abelian. They may be understood geometrically as an aristotelian limit of the Lifshitz geometries: essentially in this limit the action of the scale transformations becomes trivial. A similar interpretation can be given to the contraction $\mathfrak{a}_7 \to \mathfrak{a}_6$, where the four-dimensional Lie group $\mathcal{G}$, which is a semidirect product $\mathbb{R}^+ \ltimes \mathbb{N}$ becomes a direct product $\mathbb{R}^+ \times \mathbb{N}$ again via the trivialisation of the scale transformations.

## 5 Geometrical properties of the spacetimes

### 5.1 Invariants of Lifshitz spacetimes

The Lifshitz spacetimes in Table 2 are all homogeneous spaces of the form G/H where $H \cong SO(d)$. Weyl [36, Theorem 2.11.A] proved that the primitive tensor invariants of $SO(d)$, out of which any other invariant tensor can be written, are the Kronecker $\delta_{ab}$ and the Levi-Civita $\epsilon_{ab\cdots c}$. Therefore all homogeneous Lifshitz spacetimes in Table 2 share the same invariant tensors. In low rank, they are given by vector fields H and D, their dual one-forms $\eta, \delta$, as well as $\pi^2 = \delta_{ab}\pi^a\pi^b$ and $P^2 = \delta^{ab}P_a P_b$. In addition we have the corresponding volume forms. Notice that each of these spaces admits invariant metrics of signatures $(d+2, 0)$, $(d+1, 1)$ and $(d, 2)$; although perhaps it is the lorentzian case which is the most relevant in the present context. Even in this case, there is of course a choice: e.g., $\pi^2 + 2\eta\delta$ and $\pi^2 + \delta^2 - \eta^2$ give rise to different invariant lorentzian metrics.

We will now show that the Lifshitz metric (2) is indeed one of the invariant metrics of the homogeneous Lifshitz spacetime L3$_z$. Parametrising the group element as $\sigma(t, \boldsymbol{x}, \rho) = e^{tH + \boldsymbol{x} \cdot \boldsymbol{P}} e^{\rho D}$ we can calculate the pull back of the (left-invariant) Maurer-Cartan form $\vartheta$ (for the details of this computation see, e.g., [35, Section 3.6.])

$$\sigma^*\vartheta = e^{-z\rho}dtH + e^{-\rho}d\boldsymbol{x} \cdot \boldsymbol{P} + d\rho D = \theta. \tag{40}$$

Since the canonical invariant connection vanishes it is also equivalent to the soldering form $\theta$. We can now use the soldering form to map the invariant tensors to the tangent space of the manifold, e.g., $\theta(\eta) = e^{-z\rho}dt$, $\theta(\delta) = d\rho$, and $\theta(\pi^a) = e^{-\rho}dx^a$. We can now write

$-\eta^2 + \delta^2 + \pi^2$ in coordinates

$$-e^{-2z\rho}\mathrm{d}t^2 + \mathrm{d}\rho^2 + e^{-2\rho}\mathrm{d}\boldsymbol{x} \cdot \mathrm{d}\boldsymbol{x}, \tag{41}$$

which with the change of coordinates $r = e^\rho$ leads us to the Lifshitz metric (2).

## 5.2 Invariants of Lifshitz–Weyl spacetimes

For the $(d + 1)$-dimensional Lifshitz–Weyl spacetimes in Table 3 – that is, those $(d + 1)$-dimensional spaces corresponding to effective Klein pairs – the natural invariants are not tensors but what we could term conformal classes of tensors. These Lifshitz–Weyl spacetimes are quotients of the Lifshitz spacetimes in Table 2 by the one-parameter subgroup generated (in the notation of Table 3) by D. As homogeneous spacetimes, the Lifshitz–Weyl spacetimes are diffeomorphic to $G/H$, where $H \cong CO(d) \cong SO(d) \times \mathbb{R}^+$ is the d-dimensional similitude group. In particular, they are base manifolds for a principal H-bundle for which all the tensor bundles are associated bundles. For example, the tangent bundle of a Lifshitz–Weyl spacetime M is associated to the reducible representation of $CO(d)$ given by $(V \otimes L^\lambda) \oplus (S \otimes L^\mu)$, where V and S are the vector and scalar representations of $SO(d)$, respectively, and $L^w$ is the one-dimensional representation of the subgroup of dilatations of weight $w$. Here, $\lambda$ and $\mu$ are the D-weights of $P_a$ and H, respectively. The natural invariant tensors on M are then rotational invariants which transform according to some weight. The result of Weyl [36] quoted above says that the rotational invariant tensors of low rank are the vector H, the dual one-form $\eta$, the symmetric rank-2 covariant tensor $\pi^2$ and the symmetric bivector $P^2$. In Table 7 we tabulate the "conformal" weights of these low-rank invariant tensors for the Lifshitz–Weyl spacetimes. It follows from this table that the only such spacetime with a conformal structure (either lorentzian $\pi^2 - \eta^2$ or riemannian $\pi^2 + \eta^2$) is $LW2_{z=1}$, corresponding to conformally compactified Minkowski spacetime, since $L2_{z=1}$ is the Poincaré patch of AdS and $LW2_{z=1}$ is the quotient by the dilatations, which as mentioned in the introduction is diffeomorphic to the conformal boundary.

Table 7: Conformal weights of low-rank invariants of homogeneous Lifshitz–Weyl spacetimes.

| LW# | Weights | | | |
| --- | --- | --- | --- | --- |
| | H | $\eta$ | $P^2$ | $\pi^2$ |
| 1 | 1 | $-1$ | 0 | 0 |
| $2_z$ | $z$ | $-z$ | 2 | $-2$ |
| $3_\pm$ | 1 | $-1$ | 0 | 0 |
| 4 | 2 | $-2$ | 2 | $-2$ |

## 5.3 Invariants of the aristotelian spacetimes

Since the action of the charge Q on the geometry is not effective the invariants are the same as for the aristotelian spacetimes without the scalar charge. In particular they admit all the aforementioned invariants H, $\eta$, $P^2$ and $\pi^2$ as true invariants, not just as conformal ones.

# 6 Lifshitz particles

We now shift attention to another class of homogeneous manifolds of Lifshitz Lie groups, the Lie groups of the Lifshitz Lie algebras in Table 1. They are not to be interpreted as spacetimes, but as elementary systems (loosely, particles) with Lifshitz symmetry; that is, symplectic manifolds admitting a transitive action of a Lifshitz group via symplectomorphisms. Let us briefly review the relationship between homogeneous symplectic manifolds and coadjoint orbits of certain central extensions. A more detailed description motivated by the present paper can be found in [37].

## 6.1 Coadjoint orbits

Let G be a connected Lie group acting transitively on a simply-connected symplectic manifold $(M, \omega)$ via symplectomorphisms. If we let $g \in G$ also denote the diffeomorphism of M induced by g, then this condition is simply $g^*\omega = \omega$. As shown by Souriau [33], associated to such data there is a moment map $\mu : M \to \mathfrak{g}^*$, with $\mathfrak{g}$ the Lie algebra of G, defined up to the addition of a constant element of $\mathfrak{g}^*$ and such that it is G-equivariant: intertwining between the G-action on M and an affinisation of the coadjoint action of G on $\mathfrak{g}^*$; that is, for all $g \in G$ and $p \in M$,

$$\mu(g \cdot p) = \mathrm{Ad}_g^* \mu(p) + \theta(g), \tag{42}$$

where $\theta : G \to \mathfrak{g}^*$ is a symplectic group cocycle. In other words, $\theta$ obeys the cocycle condition

$$\theta(g_1 g_2) = \theta(g_1) + \mathrm{Ad}_{g_1}^* \theta(g_2), \tag{43}$$

and its derivative $d_e\theta : \mathfrak{g} \to \mathfrak{g}^*$ at the identity is such that

$$\langle (d_e\theta)(X), Y \rangle = - \langle (d_e\theta)(Y), X \rangle. \tag{44}$$

If $\theta$ were a coboundary, so that $\theta(g) = \mu_0 - \mathrm{Ad}_g^* \mu_0$ for some constant $\mu_0 \in \mathfrak{g}^*$, then we could redefine the moment map: $\mu \mapsto \mu' = \mu - \mu_0$, so that now

$$\mu'(g \cdot p) = \mathrm{Ad}_g^* \mu'(p) \tag{45}$$

is equivariant relative to the (linear) coadjoint action. If $\theta$ is cohomologically nontrivial, then it defines a one-dimensional central extension $\widehat{G}$ of G (see [37, Theorem 15]) and the affine action of G on $\mathfrak{g}^*$ is now essentially the coadjoint representation of $\widehat{G}$ on $\widehat{\mathfrak{g}}^*$. It then follows that M is the universal cover of a coadjoint orbit of $\widehat{G}$.

One-dimensional central extensions[4] of G are classified by the smooth group cohomology group $H^2(G)$. The celebrated van Est theorem [38] (see also [39]) implies that $H^2(G)$ is isomorphic to the relative Lie algebra cohomology $H^2(\mathfrak{g}, \mathfrak{k})$, where $\mathfrak{k}$ is the Lie algebra of a maximal compact subgroup of G. Hence to determine (up to coverings) the homogeneous symplectic manifolds of G we need to determine the coadjoint orbits of every one-dimensional central extension of G whose van Est derivative defines a class in $H^2(\mathfrak{g}, \mathfrak{k})$.

As an illustration of how coadjoint orbits arise from particle motion, let us consider briefly geodesic motion in the Lifshitz spacetime $(M^{d+2}, g)$ where g is the metric tensor in equation (2). For generic values of z (here $z \neq 0, 1$), g admits Killing vector fields given by equation (3) generating an action of the Lifshitz group G with Lie algebra $\mathfrak{g} = \mathfrak{a}_3^z$ on M by isometries. Let $\gamma : I \to M$ be an affinely-parametrised geodesic of the Levi-Civita connection defined by g. Every such geodesic $\gamma$ defines an element $\alpha_\gamma \in \mathfrak{g}^*$; that is, a linear map $\alpha_\gamma : \mathfrak{g} \to \mathbb{R}$ given by $X \mapsto g(\dot{\gamma}, \xi_X)$, which is a constant along the geodesic. If we let $a \in G$, let $\phi_a : M \to M$ denote

---

[4]Strictly speaking with kernel $\cong \mathbb{R}$, but the topology of the kernel is of no consequence.

the corresponding diffeomorphism. Then $\phi_a \circ \gamma : I \to M$ is also an affinely parametrised geodesic and it is not hard to show that $\alpha_{\phi_a \circ \gamma} = \mathrm{Ad}_a^* \alpha_\gamma$. Therefore a geodesic $\gamma$ defines a map $G \to \mathfrak{g}^*$, sending $a \in G$ to $\alpha_{\phi_a \circ \gamma} = \mathrm{Ad}_a^* \alpha_\gamma$, which is none other but the coadjoint orbit map. The image of this map is precisely the coadjoint orbit of $\alpha_\gamma$.

Let us observe that this assignment from particle trajectories in a spacetime to coadjoint orbits is not bijective, in that different spacetimes might give rise to the same coadjoint orbit. This simply reflects the fact that coadjoint orbits are an intrinsic property of the Lie group and not a property of the spacetime.

In this section we present some partial results on the calculation of coadjoint orbits of some Lifshitz Lie groups. We divide the discussion into parts labelled by the Lie algebras in Table 1.

## 6.2 $\mathfrak{a}_1$, $\mathfrak{a}_2$ and $\mathfrak{a}_3^z$

As in Section 3.5.1, we treat them together by introducing a new parameter $w$ and declaring $[D, P_a] = w P_a$. Rescaling $D$, we see that $(w, z)$ are defined only up to multiplication by a nonzero real number. The choice $(z, w) = (0, 0)$ gives $\mathfrak{a}_1$, whereas $\mathfrak{a}_2$ corresponds to $(z, w) = (1, 0)$ and $\mathfrak{a}_3^z$ corresponds to $(z, w) = (z, 1)$.

### 6.2.1 Adjoint and coadjoint actions

Let $\mathfrak{g}$ be the Lie algebra spanned by $J_{ab}, P_a, H, D$ subject to the brackets (6) together with

$$[P_a, P_b] = 0, \qquad [H, P_a] = 0, \qquad [D, P_a] = w P_a \qquad \text{and} \qquad [D, H] = z H. \qquad (46)$$

When discussing coadjoint orbits associated to a Lie algebra $\mathfrak{g}$ we need to specify the Lie group $G$ under consideration. One way is to take $G$ to be the unique connected and simply-connected group with Lie algebra $\mathfrak{g}$. This group typically does not act effectively on $\mathfrak{g}$ (and hence on its dual), but a certain quotient (known as the adjoint group) does. For example, for $\mathfrak{g} = \mathfrak{su}(n)$, one would take $G = SU(n)$ and the adjoint group is the quotient by the $\mathbb{Z}_n$ subgroup consisting of scalar matrices. For the Lie algebra $\mathfrak{g}$ under consideration, defined by the brackets (6) and (46), we may take $G$ to be the group with underlying manifold $\mathbb{R}^+ \times SO(d) \times \mathbb{R}^d \times \mathbb{R}$ and multiplication given by

$$(\sigma_1, A_1, v_1, h_1) \cdot (\sigma_2, A_2, v_2, h_2) = (\sigma_1 \sigma_2, A_1 A_2, v_1 + \sigma_1^w A_1 v_2, h_1 + \sigma_1^z h_2). \qquad (47)$$

The group $G$ is not simply-connected: its universal cover would have underlying manifold $\mathbb{R}^+ \times \mathrm{Spin}(d) \times \mathbb{R}^d \times \mathbb{R}$ (if $d > 2$) or $\mathbb{R}^+ \times \mathbb{R}^2 \times \mathbb{R}^2 \times \mathbb{R}$ (if $d = 2$), but all the representations of $\mathfrak{so}(d)$ appearing in the Lie algebra are tensorial and hence factor through $SO(d)$, so that the adjoint group is $G$, at least when $z \neq 0$. If $z = 0$ then the adjoint representation has nontrivial kernel and the adjoint group is the quotient of $G$ by this kernel.

It follows from the multiplication law (47) that the identity of $G$ is $(1, \mathbb{1}, 0, 0)$ and inversion is given by

$$(\sigma, A, v, h)^{-1} = (\sigma^{-1}, A^{-1}, -\sigma^{-w} A^{-1} v, -\sigma^{-z} h). \qquad (48)$$

As a check, let us calculate the Lie algebra of $G$. Consider a curve $\gamma(t) = (\sigma(t), A(t), v(t), h(t))$ in $G$ with $\gamma(0) = (1, \mathbb{1}, 0, 0)$ and $\gamma'(0) = (\lambda, X, p, \varepsilon) \in \mathbb{R} \oplus \mathfrak{so}(d) \oplus \mathbb{R}^d \oplus \mathbb{R}$.

The adjoint action on $g \in G$ on $\gamma'(0) \in \mathfrak{g}$ is given by the velocity at the identity of the curve $g\gamma(t)g^{-1}$. If we let $g = (\sigma, A, v, h)$ and $\gamma'(0) = (\lambda, X, q, \theta)$ as before, we find that

$$\mathrm{Ad}_{(\sigma, A, v, h)}(\lambda, X, q, \theta) = (\lambda, AXA^{-1}, \sigma^w Aq - w\lambda v - AXA^{-1}v, \sigma^z \theta - z\lambda h). \qquad (49)$$

It follows from this expression that

$$\ker \mathrm{Ad} = \begin{cases} \{(1, \mathbb{1}, 0, 0)\}, & \text{if } z \neq 0, \\ \{(1, \mathbb{1}, 0, h)|h \in \mathbb{R}\}, & \text{if } z = 0 \text{ and } w \neq 0, \\ \{(\sigma, \mathbb{1}, 0, h)|\sigma \in \mathbb{R}^+, \ h \in \mathbb{R}\}, & \text{if } z = w = 0. \end{cases} \qquad (50)$$

Replacing $(\sigma, A, \nu, h)$ by a curve $(\sigma(s), A(s), \nu(s), h(s))$ in $G$ through the identity in equation (49), we obtain a curve in $\mathfrak{g}$ through $(\lambda, X, q, \theta)$, whose velocity there is the bracket

$$[(\sigma'(0), A'(0), \nu'(0), h'(0)), (\lambda, X, q, \theta)].\tag{51}$$

Performing this calculation we obtain (after changing notation)

$$\begin{aligned}&[(\lambda_1, X_1, q_1, \theta_1), (\lambda_2, X_2, q_2, \theta_2)]\\&= (0, [X_1, X_2], w(\lambda_1 q_2 - \lambda_2 q_1) + X_1 q_2 - X_2 q_1, z(\lambda_1 \theta_2 - \lambda_2 \theta_1)).\end{aligned}\tag{52}$$

Comparing with (6) and (46), we see that

$$J_{ab} = (0, E_{ab}, 0, 0), \quad P_a = (0, 0, e_a, 0), \quad H = (0, 0, 0, 1) \quad \text{and} \quad D = (1, 0, 0, 0),\tag{53}$$

where $e_a$ are the elementary vectors in $\mathbb{R}^d$ and $E_{ab} \in \mathfrak{so}(d)$ is the skew-symmetric endomorphism defined by

$$E_{ab} e_c = \delta_{bc} e_a - \delta_{ac} e_b.\tag{54}$$

We may identify $\mathfrak{g}^*$ with $\mathfrak{g}$ as vector spaces, under the inner product $\langle -, - \rangle : \mathfrak{g} \times \mathfrak{g} \to \mathbb{R}$ defined by

$$\langle (\mu, Y, p, \varepsilon), (\lambda, X, q, \theta) \rangle = \lambda \mu + \tfrac{1}{2} \operatorname{tr}(Y^\mathsf{T} X) + p^\mathsf{T} q + \theta \varepsilon.\tag{55}$$

In this way, we can identify the canonical dual basis $\lambda^{ab}, \pi^a, \eta, \delta$ for $\mathfrak{g}^*$ with

$$\lambda^{ab} = (0, E_{ab}, 0, 0), \quad \pi^a = (0, 0, e_a, 0), \quad \eta = (0, 0, 0, 1) \quad \text{and} \quad \delta = (1, 0, 0, 0).\tag{56}$$

This inner product is not invariant under the adjoint representation, so that the adjoint and coadjoint representations are not equivalent. The coadjoint action can be worked out using the above inner product:

$$\left\langle \operatorname{Ad}^*_{(\sigma, A, \nu, h)}(\mu, Y, p, \varepsilon), (\lambda, X, q, \theta) \right\rangle = \left\langle (\mu, Y, p, \varepsilon), \operatorname{Ad}_{(\sigma, A, \nu, h)^{-1}}(\lambda, X, q, \theta) \right\rangle.\tag{57}$$

Using the explicit expression (49) for the adjoint action, we find that the coadjoint action is given by

$$\operatorname{Ad}^*_{(\sigma, A, \nu, h)}(\mu, Y, p, \varepsilon) = (\mu', Y', p', \varepsilon'),\tag{58}$$

where

$$\begin{aligned}\mu' &= \mu + w\sigma^{-w}\nu^\mathsf{T} A p + z\sigma^{-z} h \varepsilon,\\Y' &= AYA^{-1} + \sigma^{-w}(Ap\nu^\mathsf{T} - \nu(Ap)^\mathsf{T}),\\p' &= \sigma^{-w} A p,\\\varepsilon' &= \sigma^{-z} \varepsilon,\end{aligned}\tag{59}$$

which differs from the adjoint action (49), as expected. This expression is to be interpreted as the action of the group $G = \mathbb{R}^+ \times \operatorname{SO}(d) \times \mathbb{R}^d \times \mathbb{R}$ (with the group multiplication (47)) on the vector space $\mathbb{R} \oplus \wedge^2 \mathbb{R}^d \oplus \mathbb{R}^d \oplus \mathbb{R}$. Notice, parenthetically, that for nonzero $\varepsilon$ and $p$ the rational function $\frac{\varepsilon^{2/z}}{(p^\mathsf{T} p)^{1/w}}$ is an invariant of the coadjoint orbit. To interpret this invariant we restrict to $w = 1$ and rewrite it as $\varepsilon^2 = \nu_z^2 (p^\mathsf{T} p)^z$ which we can understand as a dispersion relation. For $z = 1$ it indeed agrees with the well known relation $\varepsilon^2 = c^2 p^\mathsf{T} p$ of massless Poincaré particles where $\nu_1$ is given by the speed of light $c$. This also agrees with the Lifshitz particle presented in [40]. For vanishing $p$ the spin $\operatorname{tr}(Y^\mathsf{T} Y)$ is an invariant and when additionally $\varepsilon$ is zero $\mu$ is also invariant.

Special cases of the coadjoint action are

$$
\begin{aligned}
\mathrm{Ad}^*_{(\sigma,\mathbb{1},0,0)}(\mu, Y, p, \varepsilon) &= (\mu, Y, \sigma^{-w}p, \sigma^{-z}\varepsilon)\,, \\
\mathrm{Ad}^*_{(1,A,0,0)}(\mu, Y, p, \varepsilon) &= (\mu, AYA^{-1}, Ap, \varepsilon)\,, \\
\mathrm{Ad}^*_{(1,\mathbb{1},v,0)}(\mu, Y, p, \varepsilon) &= (\mu + wv^\mathsf{T}p, Y + pv^\mathsf{T} - vp^\mathsf{T}, p, \varepsilon)\,, \\
\mathrm{Ad}^*_{(1,\mathbb{1},0,h)}(\mu, Y, p, \varepsilon) &= (\mu + zh\varepsilon, Y, p, \varepsilon)\,.
\end{aligned}
\tag{60}
$$

The first two are as expected: $(\sigma,\mathbb{1},0,0)$ rescales $p$ and $\varepsilon$, whereas $(1,A,0,0)$ acts like a rotation: conjugating $Y$ and rotating the vector $p$.

### 6.2.2 Structure of coadjoint orbits

The group $G$ is actually a semidirect product $\mathrm{CO}(d) \ltimes T$, where the abelian group $T \cong \mathbb{R}^{d+1}$ and where the action of $(\sigma, A) \in \mathrm{CO}(d)$ on $(v, h) \in T$ is given by

$$
(\sigma, A) \cdot (v, h) = (\sigma^w Av, \sigma^z h)\,.
\tag{61}
$$

Coadjoint orbits of such semidirect products have been studied, for example, in the thesis of Oblak [41] and the references therein. Let us write $G = K \ltimes T$, with $K$ a connected Lie group and $T$ abelian. We will let $\mathfrak{g} = \mathfrak{k} \oplus \mathfrak{t}$ as a vector space and hence $\mathfrak{g}^* = \mathfrak{k}^* \oplus \mathfrak{t}^*$, where we identify $\mathfrak{k}^*$ with the annihilator $\mathfrak{t}^o = \{\alpha \in \mathfrak{g}^* | \alpha(X) = 0 \ \forall X \in \mathfrak{t}\}$ of $\mathfrak{t}$ and, similarly, $\mathfrak{t}^*$ with the annihilator $\mathfrak{k}^o$ of $\mathfrak{k}$. Let $\alpha \in \mathfrak{g}^*$ and decompose it as $\alpha = (\kappa, \tau) \in \mathfrak{k}^* \oplus \mathfrak{t}^*$. Since $K$ acts on $T$ by automorphisms, it acts on its Lie algebra $\mathfrak{t}$ and hence on the dual $\mathfrak{t}^*$. Let $\mathcal{O}_\tau \subset \mathfrak{t}^*$ denote the $K$-orbit of $\tau$ in $\mathfrak{t}^*$. There is a $K$-equivariant diffeomorphism $\mathcal{O}_\tau \cong K/K_\tau$, where $K_\tau = \{k \in K | k \cdot \tau = \tau\}$ is the stabiliser of $\tau$ in $K$. This exhibits $K$ as the total space of a principal $K_\tau$ bundle $K \to \mathcal{O}_\tau$ and given any manifold $M$ on which $K_\tau$ acts, we may construct an associated fibre bundle $K \times_{K_\tau} M \to \mathcal{O}_\tau$, whose typical fibre is a copy of $M$. For example, $K_\tau$ acts on $\mathfrak{k}^*$ and, since $\mathfrak{k}_\tau \subset \mathfrak{k}$ is a Lie subalgebra, this action preserves the annihilator $\mathfrak{k}_\tau^o$. Since this is a linear representation of $K_\tau$, the associated fibre bundle $K \times_{K_\tau} \mathfrak{k}_\tau^o$ is a vector bundle, and using the isomorphism $\mathfrak{k}_\tau^o \cong (\mathfrak{k}/\mathfrak{k}_\tau)^*$ can be seen to be the cotangent bundle $T^*\mathcal{O}_\tau$ of $\mathcal{O}_\tau$. Another example of associated fibre bundles, this time not a vector bundle, is given by considering a coadjoint orbit $\mathcal{O}'$ of $K_\tau$ and constructing $K \times_{K_\tau} \mathcal{O}'$. With these definitions behind us, we can describe the coadjoint orbits of $G = K \ltimes T$. The $G$-coadjoint orbit of $\alpha = (\kappa, \tau)$ is the associated fibre bundle $K \times_{K_\tau} (\mathfrak{k}_\tau^o \times \mathcal{O}_{\kappa_\tau}) \to \mathcal{O}_\tau$, where $\kappa_\tau \in \mathfrak{k}_\tau^*$ is the restriction of $\kappa$ to $\mathfrak{k}_\tau$ and $\mathcal{O}_{\kappa_\tau}$ is its $K_\tau$-coadjoint orbit. The total space of the bundle $K \times_{K_\tau} (\mathfrak{k}_\tau^o \times \mathcal{O}_{\kappa_\tau}) \to \mathcal{O}_\tau$ is the fibred product of the cotangent bundle $T^*\mathcal{O}_\tau$ and the associated fibre bundle $K \times_{K_\tau} \mathcal{O}_{\kappa_\tau}$ over $\mathcal{O}_\tau$. As a check, notice that the dimension is $2\dim\mathcal{O}_\tau + \dim\mathcal{O}_{\kappa_\tau}$, which is indeed even, since $\mathcal{O}_{\kappa_\tau}$ is a coadjoint orbit itself.

Two extremal cases are worth noting: if $\tau = 0$, then this simply the $K$-coadjoint orbit of $\kappa$, whereas if $\kappa = 0$, this is simply the cotangent bundle $T^*\mathcal{O}_\tau$.

To determine the coadjoint orbits of our groups of interest $G = \mathrm{CO}(d) \ltimes T$ we need to first decompose $\mathfrak{t}^*$ into $\mathrm{CO}(d)$-orbits and determine their stabilisers and then to determine the coadjoint orbits of the stabilisers. The action of $\mathrm{CO}(d)$ on $\mathfrak{t}^*$ can be read off from the last two entries in equation (58) for the coadjoint action after setting $v$ and $h$ to zero:

$$
(\sigma, A) \cdot (p, \varepsilon) = (\sigma^{-w} Ap, \sigma^{-z}\varepsilon)\,.
\tag{62}
$$

To continue, we must consider several cases depending on the values of $(z, w)$.

### 6.2.3 $\mathrm{CO}(d)$-orbits in $\mathfrak{t}^*$ for $z = w = 0$

If $z = w = 0$ the action (62) reduces to

$$
(\sigma, A) \cdot (p, \varepsilon) = (Ap, \varepsilon)\,.
\tag{63}
$$

There are two kinds of orbits:

- point-like orbits $\{(0, \varepsilon)\}$, with stabiliser $CO(d)$; and

- spherical orbits $S^{d-1}_{|q|} \times \{\varepsilon\}$ through $(p \neq 0, \varepsilon)$, with stabiliser $CO(p^\perp) = \mathbb{R}^+ \times SO(p^\perp) \cong CO(d-1)$. Here the notation is that $S^{d-1}_{|p|}$ is the sphere of radius $|p|$, the euclidean norm of $p \in \mathbb{R}^d$.

### 6.2.4  $CO(d)$-orbits in $t^*$ for $w = 0$ and $z \neq 0$

We keep $w = 0$ but now have $z \neq 0$, which can be set to $z = 1$ without loss of generality by rescaling D. Then the action (62) reduces to,

$$(\sigma, A) \cdot (p, \varepsilon) = (Ap, \sigma^{-1}\varepsilon). \tag{64}$$

We have the following orbits, depending on $(p, \varepsilon)$:

- a point-like orbit $\{(0,0)\}$, with stabiliser $CO(d)$;

- a spherical orbit $S^{d-1}_{|p|} \times \{0\}$ through $(p \neq 0, 0)$, with stabiliser $CO(p^\perp)$;

- two ray-like orbits $\{0\} \times \mathbb{R}^\pm$ through $(0, \varepsilon)$ with $\pm\varepsilon > 0$ and stabilisers $SO(d)$;

- two cylindrical orbits $S^{d-1}_{|p|} \times \mathbb{R}^\pm$ through $(p \neq 0, \varepsilon)$, with $\pm\varepsilon > 0$, and stabiliser $SO(p^\perp)$.

### 6.2.5  $CO(d)$-orbits in $t^*$ for $w \neq 0$ and $z = 0$

Next we consider $z = 0$ and $w \neq 0$. Again we can set $w = 1$ without loss of generality, resulting in the action

$$(\sigma, A) \cdot (p, \varepsilon) = (\sigma^{-1}Ap, \varepsilon). \tag{65}$$

There are two kinds of orbits:

- point-like orbits $\{(0, \theta)\}$, with stabiliser $CO(d)$; and

- orbits $\left(\mathbb{R}^d \setminus \{0\}\right) \times \{\theta\}$, through $(p \neq 0, \varepsilon)$, with stabiliser $SO(p^\perp)$.

### 6.2.6  $CO(d)$-orbits in $t^*$ for $w \neq 0$ and $z \neq 0$

Finally we have the case $w = 1$ (without loss of generality) and $z \neq 0$, resulting in the action

$$(\sigma, A) \cdot (p, \varepsilon) = (\sigma^{-1}Ap, \sigma^{-z}\varepsilon). \tag{66}$$

We have the following orbits:

- a point-like orbit $\{(0,0)\}$, with stabiliser $CO(d)$;

- two ray-like orbits $\{0\} \times \mathbb{R}^\pm$ through $(0, \varepsilon)$ with $\pm\varepsilon > 0$ and stabiliser $SO(d)$;

- an orbit $\left(\mathbb{R}^d \setminus \{0\}\right) \times \{0\}$, through $(p \neq 0, 0)$, with stabiliser $SO(p^\perp)$; and

- cylindrical orbits through $(p \neq 0, \varepsilon)$ with $\pm\varepsilon > 0$, and stabiliser $SO(p^\perp)$. These orbits can be thought of as a sphere-bundle over the half-line, where the radius of the sphere varies with the point on the line, i.e., these are the generalized lightcones $\varepsilon^2 - v_z^2(p^2)^z = 0$.

### 6.2.7 Summary

In summary, the stabilisers are in all cases isomorphic to one of $CO(d)$, $CO(d-1)$, $SO(d)$ or $SO(d-1)$. We next determine the coadjoint orbits of these groups, where we restrict ourselves to $d \geqslant 2$. The coadjoint action of $CO(d)$ on $\mathfrak{co}(d)^*$ can be read off from the first two entries in equation (58) after setting $\nu$ and $h$ to zero:

$$\mathrm{Ad}^*_{(\sigma,A)}(\mu,Y) = (\mu, AYA^{-1})\,. \tag{67}$$

We therefore see that the coadjoint orbit through $(\mu,Y)$ is $\{\mu\} \times \mathcal{O}_Y$, where $\mathcal{O}_Y$ is the coadjoint orbit of $Y \in \mathfrak{so}(d)^*$ under $SO(d)$. In other words, we are left with the task of studying the coadjoint orbits of $SO(n)$ for $n \geqslant 1$, since $n = d$ or $n = d-1$ and $d \geqslant 2$. Of course, the cases $n = 1$ and $n = 2$ have only point-like orbits: this is because $\mathfrak{so}(1) = 0$ and $\mathfrak{so}(2)$ is abelian. The coadjoint orbits for $\mathfrak{so}(3)$ are well known: we have the origin of $\mathfrak{so}(3)^*$ and then the spheres of radius equal to the norm of $Y$ under the euclidean inner product induced by the Killing form. What about for $n > 3$? Being semisimple, the adjoint and coadjoint representations are equivalent, and hence we may work with the adjoint orbits. These have been characterised in [42], where it is shown in Theorem 3.1 of that paper, that coadjoint orbits of $SO(n)$ are hermitian flag manifolds in $\mathbb{R}^n$. We describe some of them in Appendix C.

   This describes all the ingredients required to determine, at least in principle, the coadjoint orbits of the Lifshitz groups associated to $\mathfrak{a}_1$, $\mathfrak{a}_2$ and $\mathfrak{a}_3^z$.

### 6.2.8 Coadjoint orbits from Lifshitz geodesics

Let us consider the case $\mathfrak{g} = \mathfrak{a}_3^z$, thought of as Killing vector fields (3) in the Lifshitz spacetime $(M^{d+2}, g)$ for the metric $g$ given in equation (2).[5] Let $\gamma(s) = (t(s), r(s), x^a(s))$ be an affinely parametrised geodesic. As discussed in Section 6.1, $\gamma$ defines an element $\alpha_\gamma = (\Delta, \ell, k, E) \in \mathfrak{g}^*$, where

$$
\begin{aligned}
\Delta &= g(\dot\gamma, \xi_D) = \frac{\dot r}{r} + z\frac{t\dot t}{r^{2z}} + \frac{x_a \dot x^a}{r^2} = \frac{\dot r}{r} - ztE + k_a x^a\,, \\
\ell_{ab} &= g(\dot\gamma, \xi_{J_{ab}}) = -x_a k_b + x_b k_a = \frac{-x_a \dot x_b + x_b \dot x_a}{r^2}\,, \\
k_a &= g(\dot\gamma, \xi_{P_a}) = \frac{\dot x_a}{r^2}\,, \\
E &= g(\dot\gamma, \xi_H) = -\frac{\dot t}{r^{2z}}\,.
\end{aligned}
\tag{68}
$$

Under the coadjoint action $\mathrm{Ad}^*_{(\sigma,A,\nu,h)}(\Delta, \ell, k, E) = (\Delta', \ell', k', E')$, where

$$
\begin{aligned}
\Delta' &= \Delta + \sigma^{-1}\nu^\mathsf{T} Ak + z\sigma^{-z}Eh\,, \\
\ell'_{ab} &= (A\ell A^{-1})_{ab} + \sigma^{-1}\left((Ak)_a \nu_b - (Ak)_b \nu_a\right)\,, \\
k'_a &= \sigma^{-1}(Ak)_a\,, \\
E' &= \sigma^{-z}E\,.
\end{aligned}
\tag{69}
$$

Consider a geodesic with $\ell = 0$ and $\Delta = 0$ and with $E \neq 0$ and $k \neq 0$. Then the corresponding coadjoint orbit has dimension $2d$ and, from the discussion in Section 6.2.2, it is the cotangent bundle $T^*\mathcal{O}_\alpha$, where $\mathcal{O}_\alpha$ is the orbit of $\alpha = k_a \pi^a + E\eta \in \mathfrak{g}^*$ under the action of the subgroup $CO(d)$ generated by $J_{ab}$ and $D$. This orbit is a generalised cylinder with equation $E^2/|k|^{2z} = c$ for some constant $c > 0$.

   The stabiliser subgroup of $\alpha = k_a \pi^a + E\eta$ is isomorphic to $SO(d-1) \times \mathbb{R}$, where $SO(d-1)$ is the subgroup of $SO(d)$ which fixes $k_a \pi^a$ and $\mathbb{R}$ is the subgroup $\Gamma \subset G$ consisting of elements of the form $(1, 1, -\frac{zEh}{|k|^2}k, h)$ for $h \in \mathbb{R}$. The coadjoint orbit of $\alpha$ is the base of a (trivial)

---

[5]There are other Lifshitz-invariant metrics, but since this section is for the purpose of illustrating the method, we pick the metric which was already discussed above.

principal $\Gamma$-bundle whose total space is the evolution space in the sense of Souriau [33]. It has a presymplectic structure (i.e., a closed 2-form) obtained by pulling back the Kirillov–Kostant–Souriau symplectic form on the coadjoint orbit, whose kernel defines a one-dimensional (integrable) distribution on the evolution space, whose leaves are the particle trajectories.

It is possible to define a particle lagrangian for such trajectories purely from the data defining the coadjoint orbit. One might argue that we already have a lagrangian, namely the one for geodesics. Let us make two remarks about this. The first is that extremals of the geodesic action principle are affinely parametrised geodesics regardless of the causal type, whereas the action constructed from the coadjoint orbit is tied to a causal type. This is well-known from the case of Minkowski geodesics, since the coadjoint orbits of lightlike and timelike geodesics are different. The second remark is that we do not always have an invariant metric on a homogeneous space and hence we do not necessarily have an action principle for geodesics, whereas we can often construct an action principle from the coadjoint orbit.

Let us illustrate this method for the coadjoint orbit of $(0, 0, k, E)$, with $k \in \mathbb{R}^d$ and $E \in \mathbb{R}$ nonzero. As we saw above the stabiliser subalgebra of $(0, 0, k, E)$ is isomorphic to $\mathfrak{so}(d-1) \oplus \mathbb{R}Z$ where $\mathfrak{so}(d-1)$ is the stabiliser of $k$ in the vector representation of $\mathfrak{so}(d)$ and $Z = (0, 0, -\frac{zE}{|k|^2}k, 1) \in \mathfrak{g}$. The corresponding evolution space is also a homogeneous space of G with stabiliser $SO(d-1)$, the subgroup of $SO(d)$ which fixes $k$. We may parametrise the evolution space locally via the coset representative

$$g = \underbrace{e^{tH}e^{x^a P_a}}_{g_0} e^{rD}e^{\theta^i R_i}, \tag{70}$$

where $R_i$, $i = 1, \ldots, d-1$ generate rotations which do not preserve $k$ and where $g_0$ parametrises a point in the $(d+1)$-dimensional Lifshitz–Weyl spacetime obtained from M by quotienting by the one-parameter group generated by D. This is just for convenience in the calculation of the pull-back of the left-invariant Maurer–Cartan one-form, which gives

$$g^{-1}dg = drD + e^{-zr}dtH + e^{-r}\left(\cos\|\theta\|dx^{\|} + \frac{\sin\|\theta\|}{\|\theta\|}\theta \cdot x^{\perp}\right)P^{\|} + \cdots, \tag{71}$$

where $x^{\|} = \frac{x \cdot k}{\|k\|^2}k$ is the component of $x$ along $k$ (and similarly for $P^{\|}$), $x^{\perp} = x - x^{\|}$ is the component perpendicular to $k$ and $\|\theta\| = \delta_{ij}\theta^i\theta^j$. In this expression we have omitted any terms which are not invariant under $\mathfrak{so}(d-1)$. If $\gamma(s)$ is a curve in the evolution space, it is described in these coordinates by $(t(s), x^a(s), r(s), \theta^i(s))$ and the lagrangian is the pull-back of a linear combination of the $\mathfrak{so}(d-1)$-invariant components of $g^{-1}dg$ to the s-interval parametrising the curve. Letting dots denote derivative with respect to $s$, we have that the most general lagrangian is given by

$$L = c_0\dot{r} + c_1 e^{-zr}\dot{t} + c_2 e^{-r}\left(\cos\|\theta\|\dot{x}^{\|} + \frac{\sin\|\theta\|}{\|\theta\|}\theta \cdot \dot{x}^{\perp}\right), \tag{72}$$

for some constants $c_0, c_1, c_2$. We can ignore the first term, since it is a total derivative, so without loss of generality we may set $c_0 = 0$.

The canonical momenta are given by

$$\begin{aligned}
E &:= \frac{\partial L}{\partial \dot{t}} = c_1 e^{-zr}, \\
p^{\|} &:= \frac{\partial L}{\partial \dot{x}^{\|}} = e^{-r}c_2 \cos\|\theta\|, \\
p_i^{\perp} &:= \frac{\partial L}{\partial \dot{x}_i^{\perp}} = e^{-r}c_2 \frac{\sin\|\theta\|}{\|\theta\|}\theta_i.
\end{aligned} \tag{73}$$

The Euler–Lagrange equations say that $E, p^\parallel, p_i^\perp$ are constant, so that if we assume that none of $z, c_1, c_2$ are zero, we obtain that $r$ and $\theta_i$ are constant. The momentum satisfies the constraint

$$p^2 = (p^\parallel)^2 + (p^\perp)^2 = c_2^2 e^{-2r}, \tag{74}$$

so that we recover the constraint that is satisfied by the coadjoint orbit: namely, $E^2/\|p\|^{2z} = (c_1/c_2^z)^2$. For prior work related to dynamical realisations of the Lifshitz group see the unpublished work of Gomis and Kamimura described in [40] and the more recent work of Galajinsky [43].

### 6.2.9 Central extensions

As shown in Table 6, for $d = 2$ the Lie algebras $\mathfrak{a}_1$ and $\mathfrak{a}_2$ admit central extensions which integrate to central extensions of the corresponding Lie groups. Coadjoint orbits of these central extensions give rise to homogeneous symplectic manifolds of the corresponding two-dimensional Lifshitz Lie groups. We shall not discuss them in this paper, but might it be interesting to study them in the future.

The Lie algebras $\mathfrak{a}_1$ and $\mathfrak{a}_3^{z=0}$ do have a central extension for all values of $d$. In the case of $\mathfrak{a}_1$, the central extension is isomorphic to $\mathfrak{iso}(d) \oplus \mathfrak{h}_3$, where $\mathfrak{h}_3$ is a Heisenberg algebra. Coadjoint orbits of a direct product of Lie groups are products of coadjoint orbits: those of the Heisenberg group are discussed in Appendix B, whereas those of the euclidean group $ISO(d) = SO(d) \ltimes \mathbb{R}^d$ can be obtained via the method explained in Section 6.2.2: they boil down to the determination of the coadjoint orbits of $SO(d-1)$, being the stabiliser of a nonzero $p \in (\mathbb{R}^d)^*$. The central extension of $\mathfrak{a}_3^{z=0}$ is now isomorphic to a semidirect product $(\mathfrak{so}(d) \oplus \mathfrak{h}_3) \ltimes \mathbb{R}^d$, where $\mathbb{R}^d = \langle P_a \rangle$ is an abelian ideal. The discussion in Section 6.2.2 again applies and all the situation is very similar to the one described above with $SO(d) \times H_3$ replacing $CO(d)$. We do not discuss them further here, but leave them for future work.

## 6.3 $\mathfrak{a}_4^\pm$ and $\mathfrak{a}_5^\pm$

We discuss these two Lie algebras together by introducing a parameter $z \in \{0, 1\}$ and letting $[D, H] = zH$. The Lie algebra $\mathfrak{a}_4^\pm$ corresponds to $z = 0$ and $\mathfrak{a}_5^\pm$ to $z = 1$. These Lie algebras are direct sums of the Lie algebras: the subalgebra spanned by $J_{ab}, P_a$ and the two-dimensional Lie algebra spanned by $D, H$. The former Lie algebra has brackets $[P_a, P_b] = \pm J_{ab}$ in addition to those involving $J_{ab}$. It is isomorphic to $\mathfrak{so}(d, 1)$ if the sign is $+$ and to $\mathfrak{so}(d+1)$ if the sign is $-$. A coadjoint orbit of either of these two Lifshitz Lie groups is therefore a product of a coadjoint orbit of $SO(d, 1)_0$ or $SO(d+1)$, for $d \geqslant 2$ and a coadjoint orbit of the two-dimensional Lie group generated by $D$ and $H$. They are described in the appendices.

The connected (and simply-connected) Lie group $G$ generated by $D$ and $H$ is given by

$$G = \left\{ \begin{pmatrix} a & b \\ 0 & a^{1-z} \end{pmatrix} \middle| a, b \in \mathbb{R}, \ a > 0 \right\}. \tag{75}$$

Let us consider $\mathfrak{a}_5^\pm$, so $z = 1$. As shown in Section 3.5.2, this Lie algebra admits no central extensions and hence any simply-connected homogeneous symplectic manifold covers a coadjoint orbit. These are determined in Appendix A.

As shown in Section 3.5.2, the Lie algebra $\mathfrak{a}_4^\pm$ admits a one-dimensional central extension $[D, H] = Z$. This promotes the two-dimensional abelian group generated by $D, H$ to the Heisenberg group, whose coadjoint orbits are described in Appendix B. The relevant coadjoint orbits of the central extension are now products of coadjoint orbits of the Heisenberg group with those of either $SO(d+1)$ or $SO(d, 1)_0$.

## 6.4 $\mathfrak{a}_6$ and $\mathfrak{a}_7$

As in Section 3.5.3, we introduce a parameter $w \in \{0, 1\}$ in order to treat both algebras simultaneously. Let $\mathfrak{g}$ (depending on $w$) be the Lie algebra under consideration. The brackets are then

$$[D, P_a] = wP_a\,, \qquad [D, H] = 2wH\,, \qquad [J, P_a] = \epsilon_{ab}P_b \qquad \text{and} \qquad [P_a, P_b] = \epsilon_{ab}H\,. \quad (76)$$

It has the structure of a semidirect product of the abelian subalgebra spanned by $J$ and $D$ acting as derivations on the Heisenberg ideal spanned by $P_a, H$.

The Lie algebra $\mathfrak{g} = \mathfrak{k} \ltimes \mathfrak{h}_3$ is a semidirect product of the abelian two-dimensional Lie algebra $\mathfrak{k} = \langle J, D \rangle \cong \mathfrak{co}(2)$ with the Heisenberg algebra: $\mathfrak{h}_3 = \langle P_a, H \rangle$.

### 6.4.1 Group law and (co)adjoint actions

The first task is to explicitly write down the group law in $G = K \ltimes H_3$. Let $H_3$ be the Heisenberg group generated by $P_a, H$. It is a unipotent matrix group diffeomorphic to $\mathbb{R}^3$, given explicitly by

$$H_3 = \left\{ \begin{pmatrix} 1 & a & c \\ 0 & 1 & b \\ 0 & 0 & 1 \end{pmatrix} \middle| a, b, c \in \mathbb{R} \right\}\,. \quad (77)$$

Every element of $H_3$ can be uniquely written as a product of matrix exponentials:

$$\begin{pmatrix} 1 & a & c \\ 0 & 1 & b \\ 0 & 0 & 1 \end{pmatrix} = \exp \begin{pmatrix} 0 & 0 & c \\ 0 & 0 & 0 \\ 0 & 0 & 0 \end{pmatrix} \exp \begin{pmatrix} 0 & 0 & 0 \\ 0 & 0 & b \\ 0 & 0 & 0 \end{pmatrix} \exp \begin{pmatrix} 0 & a & 0 \\ 0 & 0 & 0 \\ 0 & 0 & 0 \end{pmatrix} \quad (78)$$
$$= \exp(cH)\exp(bP_2)\exp(aP_1)\,,$$

where

$$P_1 = \begin{pmatrix} 0 & 1 & 0 \\ 0 & 0 & 0 \\ 0 & 0 & 0 \end{pmatrix}\,, \qquad P_2 = \begin{pmatrix} 0 & 0 & 0 \\ 0 & 0 & 1 \\ 0 & 0 & 0 \end{pmatrix} \qquad \text{and} \qquad H = \begin{pmatrix} 0 & 0 & 1 \\ 0 & 0 & 0 \\ 0 & 0 & 0 \end{pmatrix}\,. \quad (79)$$

It is easy to write down the action of $K$ on the Lie algebra $\mathfrak{h}_3$ spanned by $P_a, H$ by exponentiating $[J, P_a] = \epsilon_{ab}P_b$, $[D, P_a] = wP_a$ and $[D, H] = 2wH$. If we write $P = P_1 + iP_2$ and then the adjoint action of $J$ is simply multiplication by $-i$. Therefore we find that, of course, $\exp(\theta J)H = H$ and that

$$\exp(\theta J)(P_1 + iP_2) = e^{-i\theta}(P_1 + iP_2) = (P_1 \cos\theta + P_2 \sin\theta) + i(P_2 \cos\theta - P_1 \sin\theta)\,, \quad (80)$$

whereas

$$\exp(\sigma D)P_a = e^{w\sigma}P_a \qquad \text{and} \qquad \exp(\sigma D)H = e^{2w\sigma}H\,. \quad (81)$$

The general element $k = \exp(\sigma D)\exp(\theta J) \in K$ acts on $\mathfrak{h}_3$ as

$$\begin{aligned} g \cdot P_1 &= e^{w\sigma}(P_1 \cos\theta + P_2 \sin\theta)\,, \\ g \cdot P_2 &= e^{w\sigma}(P_2 \cos\theta - P_1 \sin\theta)\,, \\ g \cdot H &= e^{2w\sigma}H\,. \end{aligned} \quad (82)$$

One checks that this is an automorphism of $\mathfrak{h}_3$:

$$[g \cdot X, g \cdot Y] = g \cdot [X, Y]\,, \qquad \forall X, Y \in \mathfrak{h}_2\,, \quad (83)$$

so that it defines a map $\phi : K \to \text{Aut}(\mathfrak{h}_3)$ sending every $g \in K$ to the automorphism $\phi_g$ of $\mathfrak{h}_3$ defined by $\phi_g X = g \cdot X$ in equation (82). Fix $g \in K$. Then $\phi_g : \mathfrak{h}_3 \to \mathfrak{h}_3$ is in particular a Lie algebra homomorphism and thus, by the Lie correspondence, lifts to a unique Lie group homomorphism $\Phi_g : H_3 \to H_3$. To work it out we argue as follows. The graph $\gamma \subset \mathfrak{h}_3 \oplus \mathfrak{h}_3$ of $\phi_g$ is a Lie subalgebra of $\mathfrak{h}_3 \oplus \mathfrak{h}_3$ (because $\phi_g$ is a homomorphism) and hence it exponentiates there to a connected subgroup $\Gamma \subset H_3 \times H_3$. There are two cartesian projections $H_3 \times H_3 \to H_3$: restricting the left projection to $\Gamma$ gives a covering $\Gamma \to H_3$ which can be shown in this case to be an isomorphism, whereas restricting the right projection to $\Gamma$ gives the desired $\Phi_g$. In detail, the graph $\gamma \subset \mathfrak{h}_3 \oplus \mathfrak{h}_3$ is spanned by

$$
\begin{aligned}
(P_1, g \cdot P_1) &= (P_1, e^{w\sigma}(P_1 \cos \theta + P_2 \sin \theta)), \\
(P_2, g \cdot P_2) &= (P_2, e^{w\sigma}(P_2 \cos \theta - P_1 \sin \theta)), \\
(H, g \cdot H) &= (H, e^{2w\sigma} H).
\end{aligned}
\tag{84}
$$

We can write $\gamma$ explicitly as the span of the following three block-diagonal $6 \times 6$ matrices:

$$
\widehat{P}_1 := \begin{pmatrix}
0 & 1 & 0 & & & \\
0 & 0 & 0 & & & \\
0 & 0 & 0 & & & \\
& & & 0 & e^{w\sigma} \cos \theta & 0 \\
& & & 0 & 0 & e^{w\sigma} \sin \theta \\
& & & 0 & 0 & 0
\end{pmatrix},
$$

$$
\widehat{P}_2 := \begin{pmatrix}
0 & 0 & 0 & & & \\
0 & 0 & 1 & & & \\
0 & 0 & 0 & & & \\
& & & 0 & -e^{w\sigma} \sin \theta & 0 \\
& & & 0 & 0 & e^{w\sigma} \cos \theta \\
& & & 0 & 0 & 0
\end{pmatrix},
\tag{85}
$$

$$
\widehat{H} := \begin{pmatrix}
0 & 0 & 1 & & & \\
0 & 0 & 0 & & & \\
0 & 0 & 0 & & & \\
& & & 0 & 0 & e^{2w\sigma} \\
& & & 0 & 0 & 0 \\
& & & 0 & 0 & 0
\end{pmatrix},
$$

and they exponentiate to the subgroup of $\text{GL}(6, \mathbb{R})$ consisting of matrices of the form

$$
\begin{aligned}
&\exp(h\widehat{H}) \exp(p_2 \widehat{P}_2) \exp(p_1 \widehat{P}_1) \\
&= \begin{pmatrix}
1 & p_1 & h & & & \\
0 & 1 & p_2 & & & \\
0 & 0 & 1 & & & \\
& & & 1 & e^{w\sigma}(p_1 \cos \theta - p_2 \sin \theta) & e^{2w\sigma}(h - p_1 p_2 \sin^2 \theta + \frac{1}{2}(p_1^2 - p_2^2) \sin \theta \cos \theta) \\
& & & 0 & 1 & e^{w\sigma}(p_2 \cos \theta + p_1 \sin \theta) \\
& & & 0 & 0 & 1
\end{pmatrix},
\end{aligned}
\tag{86}
$$

from where read off that the Lie group automorphism $\Phi_g : H_3 \to H_3$ corresponds to

$$
\begin{pmatrix}
1 & p_1 & h \\
0 & 1 & p_2 \\
0 & 0 & 1
\end{pmatrix} \mapsto \begin{pmatrix}
1 & e^{w\sigma}(p_1 \cos \theta - p_2 \sin \theta) & e^{2w\sigma}(h - p_1 p_2 \sin^2 \theta + \frac{1}{2}(p_1^2 - p_2^2) \sin \theta \cos \theta) \\
0 & 1 & e^{w\sigma}(p_2 \cos \theta + p_1 \sin \theta) \\
0 & 0 & 1
\end{pmatrix}.
\tag{87}
$$

One checks that $\Phi : K \to \text{Aut}(H_3)$ is indeed a Lie group homomorphism, so that $\Phi_{gg'} = \Phi_g \Phi_{g'}$ for all $g, g' \in K$.

With these results in hand, we can now write down the group law on $G = K \ltimes H_3$. If $(k_1, h_1), (k_2, h_2) \in G$, their product is given by

$$(k_1, h_1)(k_2, h_2) = (k_1 k_2, h_1(\Phi_{k_1} h_2)), \tag{88}$$

from where we read off that the inverse of $(k, h) \in G$ is given by

$$(k, h)^{-1} = (k^{-1}, \Phi_{k^{-1}} h^{-1}). \tag{89}$$

Notice that any automorphism commutes with inversion so that $(\Phi_k h)^{-1} = \Phi_k h^{-1}$. Explicitly, the inverse of the group element with coordinates $(\theta, \sigma, p_1, p_2, h)$ is the group element with coordinates

$$
\begin{aligned}
\Big( -\theta, -\sigma, -e^{-w\sigma}(p_1 \cos\theta + p_2 \sin\theta), \ e^{-w\sigma}(p_1 \sin\theta - p_2 \cos\theta) \\
e^{-2w\sigma}(p_1 p_2 \cos^2\theta - h - \tfrac{1}{2}(p_1^2 - p_2^2)\sin\theta \cos\theta) \Big).
\end{aligned}
\tag{90}
$$

We can now work out the adjoint representation of $G$. If $(k, h) \in G$ and $(k_2(t), h_2(t))$ is a curve in $G$ with $k_2(0) = 1_K$, $h_2(0) = 1_{H_3}$, $k_2'(0) = X \in \mathfrak{k}$ and $h_2'(0) = Y \in \mathfrak{h}_3$, then

$$
\begin{aligned}
\widehat{\mathrm{Ad}}_{(k,h)}(X, Y) &= \left.\frac{d}{dt}\right|_{t=0} (k, h)(k_2(t), h_2(t))(k^{-1}, \Phi_{k^{-1}} h^{-1}) \\
&= \left.\frac{d}{dt}\right|_{t=0} (kk_2(t), h(\Phi_k h_2(t)))(k^{-1}, \Phi_{k^{-1}} h^{-1}) \\
&= \left.\frac{d}{dt}\right|_{t=0} (kk_2(t)k^{-1}, h(\Phi_k h_2(t))\Phi_{kk_2(t)}(\Phi_{k^{-1}} h^{-1})) \\
&= \left( \mathrm{Ad}_k^K X, \left.\frac{d}{dt}\right|_{t=0} \left( h(\Phi_k h_2(t))\Phi_{kk_2(t)k^{-1}} h^{-1} \right) \right) \\
&= \left( X, \left.\frac{d}{dt}\right|_{t=0} \left( h(\Phi_k h_2(t))\Phi_{kk_2(t)k^{-1}} h^{-1} \right) \right),
\end{aligned}
\tag{91}
$$

using that $K$ is abelian.

If we let $k = (\theta, \sigma) \in K$ and $X = (x, y) \in \mathfrak{k}$, and also

$$
h = \begin{pmatrix} 1 & p_1 & h \\ 0 & 1 & p_2 \\ 0 & 0 & 1 \end{pmatrix} \in H_3 \qquad \text{and} \qquad Y = \begin{pmatrix} 0 & u & s \\ 0 & 0 & v \\ 0 & 0 & 0 \end{pmatrix} \in \mathfrak{h}_3,
\tag{92}
$$

then $\widehat{\mathrm{Ad}}_{(k,h)}(X, Y) = (X, Y')$, where

$$
Y' = \begin{pmatrix} 0 & -wyp_1 + xp_2 + e^{w\sigma}(u\cos\theta - v\sin\theta) & \Xi \\ 0 & 0 & -wyp_2 - p_1 x + e^{w\sigma}(u\sin\theta + v\cos\theta) \\ 0 & 0 & 0 \end{pmatrix}, \tag{93}
$$

where

$$\Xi = wy(p_1 p_2 - 2h) - \tfrac{1}{2}(p_1^2 + p_2^2)x + e^{2w\sigma}s + e^{w\sigma}((up_1 + vp_2)\sin\theta + (vp_1 - up_2)\cos\theta). \tag{94}$$

Relative to the basis $(J, D, P_1, P_2, H)$ for $\mathfrak{g}$, the matrix of $\widehat{\mathrm{Ad}}_{(k,h)}$ is given by

$$
\widehat{\mathrm{Ad}}_{(k,h)} = \begin{pmatrix}
1 & 0 & 0 & 0 & 0 \\
0 & 1 & 0 & 0 & 0 \\
p_2 & -wp_1 & e^{w\sigma}\cos\theta & -e^{w\sigma}\sin\theta & 0 \\
-p_1 & -wp_2 & e^{w\sigma}\sin\theta & e^{w\sigma}\cos\theta & 0 \\
-\tfrac{1}{2}(p_1^2 + p_2^2) & w(p_1 p_2 - 2h) & e^{w\sigma}(p_1 \sin\theta - p_2 \cos\theta) & e^{w\sigma}(p_2 \sin\theta + p_1 \cos\theta) & e^{2w\sigma}
\end{pmatrix}, \tag{95}
$$

and hence relative to the canonical dual basis $(\lambda, \delta, \pi^1, \pi^2, \eta)$ for $\mathfrak{g}^*$, the matrix of the coadjoint action $\widehat{\mathrm{Ad}}^*_{(k,h)}$ is given by the inverse transpose of the above matrix:

$$\widehat{\mathrm{Ad}}^*_{(k,h)} = \begin{pmatrix} 1 & 0 & e^{-w\sigma}(p_1 \sin\theta - p_2 \cos\theta) & e^{-w\sigma}(p_1 \cos\theta + p_2 \sin\theta) & -\frac{1}{2}e^{-2w\sigma}(p_1^2 + p_2^2) \\ 0 & 1 & e^{-w\sigma}w(p_1 \cos\theta + p_2 \sin\theta) & e^{-w\sigma}w(p_2 \cos\theta - p_1 \sin\theta) & e^{-2w\sigma}w(2h - p_1 p_2) \\ 0 & 0 & e^{-w\sigma} \cos\theta & -e^{-w\sigma} \sin\theta & e^{-2w\sigma}p_2 \\ 0 & 0 & e^{-w\sigma} \sin\theta & e^{-w\sigma} \cos\theta & -e^{-2w\sigma}p_1 \\ 0 & 0 & 0 & 0 & e^{-2w\sigma} \end{pmatrix}, \quad (96)$$

so that the coadjoint action by $(k, h)$ given in equation (92) on the dual basis is explicitly:

$$\lambda \mapsto \lambda,$$
$$\delta \mapsto \delta,$$
$$\pi^1 \mapsto e^{-w\sigma}\left(\pi^1 \cos\theta + \pi^2 \sin\theta + (p_1 \sin\theta - p_2 \cos\theta)\lambda + w(p_1 \cos\theta + p_2 \sin\theta)\delta\right), \quad (97)$$
$$\pi^2 \mapsto e^{-w\sigma}\left(-\pi^1 \sin\theta + \pi^2 \cos\theta + (p_1 \cos\theta + p_2 \sin\theta)\lambda + w(p_2 \cos\theta - p_1 \sin\theta)\delta\right),$$
$$\eta \mapsto e^{-2w\sigma}\left(\eta - p_1\pi^2 + p_2\pi^1 - \tfrac{1}{2}(p_1^2 + p_2^2)\lambda + w(2h - p_1 p_2)\delta\right).$$

Let us introduce coordinates $x_\lambda, x_\delta, x_{\pi^1}, x_{\pi^2}, x_\eta$ for $\mathfrak{g}^*$. Then the infinitesimal generators of the coadjoint representation are the following vector fields on $\mathfrak{g}^*$:

$$\xi_J = -x_{\pi^2} \frac{\partial}{\partial x_{\pi^1}} + x_{\pi^1} \frac{\partial}{\partial x_{\pi^2}},$$

$$\xi_D = -wx_{\pi^1} \frac{\partial}{\partial x_{\pi^1}} - wx_{\pi^2} \frac{\partial}{\partial x_{\pi^2}} - 2wx_\eta \frac{\partial}{\partial x_\eta},$$

$$\xi_{P_1} = x_{\pi^2} \frac{\partial}{\partial x_\lambda} + wx_{\pi^1} \frac{\partial}{\partial x_\delta} - x_\eta \frac{\partial}{\partial x_{\pi^2}}, \quad (98)$$

$$\xi_{P_2} = -x_{\pi^1} \frac{\partial}{\partial x_\lambda} + wx_{\pi^2} \frac{\partial}{\partial x_\delta} + x_\eta \frac{\partial}{\partial x_{\pi^1}},$$

$$\xi_H = 2wx_\eta \frac{\partial}{\partial x_\delta}.$$

One can check that $\xi : \mathfrak{g} \to \mathscr{X}(\mathfrak{g}^*)$ given by $X \mapsto \xi_X$ is a Lie algebra antihomomorphism, as expected.

### 6.4.2 Coadjoint orbits for $w = 0$

Let us set $w = 0$. The coadjoint action on the coordinates $(x_\lambda, x_\delta, x_{\pi^1}, x_{\pi^2}, x_\eta)$ is then given by

$$x_\lambda \mapsto x_\lambda + (p_1 \sin\theta - p_2 \cos\theta)x_{\pi^1} + (p_1 \cos\theta + p_2 \sin\theta)x_{\pi^2} - \tfrac{1}{2}(p_1^2 + p_2^2)x_\eta,$$
$$x_\delta \mapsto x_\delta,$$
$$x_{\pi^1} \mapsto \cos\theta x_{\pi^1} - \sin\theta x_{\pi^2} + p_2 x_\eta, \quad (99)$$
$$x_{\pi^2} \mapsto \cos\theta x_{\pi^2} + \sin\theta x_{\pi^1} - p_1 x_\eta,$$
$$x_\eta \mapsto x_\eta.$$

We must distinguish between two cases, depending on whether or not $x_\eta$, which is inert, vanishes.

1. If $x_\eta = 0$, the orbit is either

   (a) a two-dimensional cylinder in the affine 3-plane defined by giving a constant value to $x_\delta$ and setting $x_\eta$ to zero, if at least one of $x_{\pi^1}$ and $x_{\pi^2}$ is nonzero; or

   (b) a point with coordinates $(x_\lambda, x_\delta, 0, 0, 0)$ if $x_{\pi^1} = x_{\pi^2} = 0$.

2. If $x_\eta \neq 0$, we have a two-dimensional surface in the affine 3-plane with constant $(x_\delta, x_\eta \neq 0)$, which is obtained as the graph $x_\lambda = f(x_{\pi^1}, x_{\pi^2})$ of a function in the $(x_{\pi^1}, x_{\pi^2})$-plane and hence diffeomorphic to $\mathbb{R}^2$.

### 6.4.3 Coadjoint orbits for $w = 1$

Now let's consider $w = 1$. The coadjoint action on the coordinates is now

$$
\begin{aligned}
x_\lambda &\mapsto x_\lambda + (p_1 \sin\theta - p_2 \cos\theta)e^{-\sigma}x_{\pi^1} + (p_1 \cos\theta + p_2 \sin\theta)e^{-\sigma}x_{\pi^2} - \tfrac{1}{2}(p_1^2 + p_2^2)e^{-2\sigma}x_\eta \,, \\
x_\delta &\mapsto x_\delta + (p_1 \cos\theta + p_2 \sin\theta)e^{-\sigma}x_{\pi^1} + (p_2 \cos\theta - p_1 \sin\theta)e^{-\sigma}x_{\pi^2} + (2h - p_1 p_2)e^{-2\sigma}x_\eta \,, \\
x_{\pi^1} &\mapsto e^{-\sigma}\cos\theta x_{\pi^1} - e^{-\sigma}\sin\theta x_{\pi^2} + p_2 e^{-2\sigma}x_\eta \,, \\
x_{\pi^1} &\mapsto e^{-\sigma}\sin\theta x_{\pi^1} + e^{-\sigma}\cos\theta x_{\pi^2} - p_1 e^{-2\sigma}x_\eta \,, \\
x_\eta &\mapsto e^{-2\sigma}x_\eta \,.
\end{aligned}
\tag{100}
$$

We must again distinguish between two cases:

1. if $x_\eta = 0$, the orbits are either

    (a) points with coordinates $(x_\lambda, x_\delta, 0, 0, 0)$, if $(x_{\pi^1}, x_{\pi^2}) = (0, 0)$; or

    (b) if $(x_{\pi^1}, x_{\pi^2}) \neq (0, 0)$, the orbit is four-dimensional and consists of the 4-plane $x_\eta = 0$ with the 2-plane with additional equations $x_{\pi^1} = x_{\pi^2} = 0$ removed. So diffeomorphic to $\mathbb{R}^4 \setminus \mathbb{R}^2 \cong \mathbb{R}^2 \times (\mathbb{R}^2 \setminus \{(0,0)\})$.

2. if $x_\eta \neq 0$, the orbit is a four-dimensional hypersurface given by the graph of a function of the four coordinates $(x_\lambda, x_\delta, x_{\pi^1}, x_{\pi^2})$.

## 7 Conclusion

This work provides the first systematic classification of Lifshitz algebras, spacetimes and particles. We also provide a full classification of aristotelian spacetimes with scalar charge, in particular ones with exotic spacetime symmetries. The Lifshitz algebras are summarised in Table 1 and the respective spacetimes fall into three classes: $(d + 2)$-dimensional Lifshitz spacetimes (Table 2) where the dilatations provide an additional holographic direction and the $(d + 1)$-dimensional Lifshitz–Weyl spacetimes (Table 3) and $(d + 1)$-dimensional aristotelian spacetimes with a scalar charge (Table 4). It is interesting to note that $\mathfrak{a}_2$, $\mathfrak{a}_3^{z \neq 0}$ and $\mathfrak{a}_5^{\pm}$ give rise to each of the above discussed spacetimes and can therefore be interpreted from various different angles. We refer to Section 2 for a summary of our results and end with a few remarks.

**Beyond the standard Lifshitz symmetries** Our classification was motivated by the standard Lifshitz symmetries and it is reassuring that we indeed recover them as spacetimes $\mathsf{L3}_z$ and $\mathsf{LW2}_z$. Beyond this case let us highlight the pairs based on $\mathfrak{a}_2$ (L2 and LW1) and its curved generalisation $\mathfrak{a}_5^{\pm}$ (L5 and LW3$_\pm$) both of which exist in generic dimension. It might well be that these spaces have played a rôle in the literature, if so we are unfortunately unaware of it.

**Exotic spacetime symmetries** Exotic spacetime symmetries, similar to the ones discussed in Section 2 have recently played a rôle in relation to fractons (see, e.g., [44–46] for reviews and [47–50] for earlier related work on polynomial shift symmetries). In the case of fractons the underlying geometry is also aristotelian [51, 52] and there is also an action of the spacetime symmetry on the charges. There has been a systematic study of exotic symmetries of this type [53], however the symmetries we discuss here are a generalisation of the multipole algebras [53]. We also allow for nontrivial commutation relations between the temporal, rather than just the spatial, translations and the underlying aristotelian geometry is not necessarily restricted to be flat. It is this generalisation that leads to the novel classification of exotic symmetries of Table 4. Another consequence

of our classification is that there are no aristotelian spacetimes with one scalar charge and nonzero $[Q, \mathbf{P}]$ commutator, like for the more conventional multipole algebras. With regard to [53] we have kept the full rotational symmetry, which is an assumption that might be dropped and would lead to a more general classification.

**Exotic aristotelian theories with scalar charge** By showing that these symmetries exist and can be consistently realised we have provided the first nontrivial step for the construction of exotic aristotelian theories with scalar charge. However, to further clarify their physical relevance and the relation to fracton-like theories it would be interesting to construct field theories that realise these symmetries (e.g., following [54]).

**Generalisation** In this work the spacetimes have no boost symmetry so a natural generalisation is to add an additional vector to our classification. This leads for example to Bargmann spacetimes and will be discussed in a future work [55].

The case of aristotelian geometries with one vector charge and no additional scalar has already been worked out in [30], the interpretation as exotic charges had however not been appreciated at that point.

**Unitary irreducible representations** Upon quantisation one expects that coadjoint orbits lead to unitary irreducible representations. In this sense our classification of the coadjoint orbits of the Lifshitz groups lays the foundation for such an endeavour and it might be interesting to understand if and in which sense these representations can be understood as quantum Lifshitz particles.

## Acknowledgments

We are grateful to Leo Bidussi, Joaquim Gomis, Kevin Grosvenor, Jelle Hartong, Emil Have, Yuji Hirono, Kristan Jensen, Jørgen Musaeus, Blagoje Oblak and Alfredo Pérez for useful discussions. We also thank Anton Galajinsky for bringing [43] and Mohammad Reza Mohammadi Mozaffar for bringing [24] and other interesting works to our attention.

During the start of this project the research of SP was supported by the ERC Advanced Grant "High-Spin-Grav" and by FNRS-Belgium (Convention FRFC PDR T.1025.14 and Convention IISN 4.4503.15). SP was supported by the Leverhulme Trust Research Project Grant (RPG-2019-218) "What is Non-Relativistic Quantum Gravity and is it Holographic?".

SP acknowledges support of the Erwin Schrödinger Institute (ESI) in Vienna where part of this work was conducted during the thematic programme "Geometry for Higher Spin Gravity: Conformal Structures, PDEs, and Q-manifolds".

## A  Coadjoint orbits of the two-dimensional nonabelian Lie group

The connected (and simply-connected) Lie group $\mathbb{G}$ whose Lie algebra $\mathfrak{g} = \langle D, H \rangle$ with bracket $[D, H] = H$ is isomorphic to the matrix group

$$\mathbb{G} = \left\{ \begin{pmatrix} a & b \\ 0 & 1 \end{pmatrix} \middle| a, b \in \mathbb{R}, \ a > 0 \right\},\tag{101}$$

with Lie algebra

$$\mathfrak{g} = \left\{ \begin{pmatrix} x & y \\ 0 & 0 \end{pmatrix} \middle| x, y \in \mathbb{R} \right\}.\tag{102}$$

We introduce the (non-invariant) inner product $\langle -, - \rangle$ on $\mathfrak{g}$ by

$$\langle X, Y \rangle = \operatorname{tr} X^{\mathsf{T}} Y. \tag{103}$$

This inner product defines a musical (vector space) isomorphism $\flat : \mathfrak{g} \to \mathfrak{g}^*$. If $X^\flat \in \mathfrak{g}^*$ and $g \in \mathbb{G}$ we define the coadjoint action by

$$(\operatorname{Ad}_g^* X^\flat)(Y) = X^\flat(\operatorname{Ad}_{g^{-1}} Y) = \operatorname{tr} X^{\mathsf{T}} g^{-1} Y g = \operatorname{tr} g X^{\mathsf{T}} g^{-1} Y. \tag{104}$$

Explicitly, if

$$X^{\mathsf{T}} = \begin{pmatrix} \alpha & 0 \\ \beta & 0 \end{pmatrix}, \tag{105}$$

then under the coadjoint action of $g = \begin{pmatrix} a & b \\ 0 & 1 \end{pmatrix}$,

$$\alpha \mapsto \alpha + b a^{-1} \beta \qquad \text{and} \qquad \beta \mapsto \beta a^{-1}. \tag{106}$$

Therefore the orbit of $(\alpha, \beta) = (\alpha, 0)$ is a point, whereas the orbit of $(\alpha, \beta > 0)$ is the half-plane $\beta > 0$ and that of $(\alpha, \beta < 0)$ is the half-plane $\beta < 0$.

## B Coadjoint orbits of the Heisenberg group

The Heisenberg group $\mathbb{N}$ is the three-dimensional Lie group of strictly upper triangular unipotent $3 \times 3$ matrices

$$\mathbb{N} = \left\{ \begin{pmatrix} 1 & a & c \\ 0 & 1 & b \\ 0 & 0 & 1 \end{pmatrix} \middle| a, b, c \in \mathbb{R} \right\}, \tag{107}$$

with Lie algebra

$$\mathfrak{n} = \left\{ \begin{pmatrix} 0 & x & z \\ 0 & 0 & y \\ 0 & 0 & 0 \end{pmatrix} \middle| x, y, z \in \mathbb{R} \right\}. \tag{108}$$

It is straightforward to work out the adjoint action of $\mathbb{N}$ on $\mathfrak{n}$: if $g = \begin{pmatrix} 1 & a & c \\ 0 & 1 & b \\ 0 & 0 & 1 \end{pmatrix}$,

$$\operatorname{Ad}_g \begin{pmatrix} 0 & x & z \\ 0 & 0 & y \\ 0 & 0 & 0 \end{pmatrix} = \begin{pmatrix} 0 & x & z + ay - bx \\ 0 & 0 & y \\ 0 & 0 & 0 \end{pmatrix}. \tag{109}$$

We can identify $\mathfrak{n}^*$ with $\mathfrak{n}$ as vector spaces under the musical isomorphism of the non-invariant inner product $\langle -, - \rangle$ on $\mathfrak{h}$ given by $\langle X, Y \rangle = \operatorname{tr} X^{\mathsf{T}} Y$. Then under the same element $g$ as before, the coadjoint action is given by

$$\operatorname{Ad}_g^* \begin{pmatrix} 0 & \alpha & \gamma \\ 0 & 0 & \beta \\ 0 & 0 & 0 \end{pmatrix} = \begin{pmatrix} 0 & \alpha + b\gamma & \gamma \\ 0 & 0 & \beta - a\gamma \\ 0 & 0 & 0 \end{pmatrix}. \tag{110}$$

Therefore there are two kinds of coadjoint orbits:

- the orbits of $(\alpha, \beta, \gamma) = (\alpha, \beta, 0)$, which are points; and

- the orbits of $(\alpha, \beta, \gamma \neq 0)$, which are the affine (hyper)planes $\{(\alpha, \beta, \gamma) | \alpha, \beta \in \mathbb{R}\}$.

## C  Coadjoint orbits of $SO(n)$

In this appendix we describe in some more detail the adjoint orbits of $SO(n)$.

Let $\Lambda \in \mathfrak{so}(n)$. Since $\Lambda$ is skew-symmetric, $i\Lambda$ is a hermitian endomorphism of $\mathbb{C}^n$ and thus has real eigenvalues. This means $\Lambda$ has purely imaginary eigenvalues and since $\Lambda$ is real, they are either zero or else come in complex conjugate pairs $(i\lambda, -i\lambda)$. Let $\lambda \neq 0$ and $E_{i\lambda} \subset \mathbb{C}^n$ be the eigenspace of $\Lambda$ with eigenvalue $i\lambda$. If $v \in E_{i\lambda}$, its complex conjugate $\bar{v} \in E_{-i\lambda}$. Write $v = x + iy$, where $x, y \in \mathbb{R}^n$. Then $\Lambda x = -\lambda y$ and $\Lambda y = \lambda x$. If $i\lambda$ has multiplicity $m$, then we can choose a unitary basis $v_1, \dots, v_m$ for $E_{i\lambda}$ and writing $v_j = x_i + iy_j$, we have that on the span of $x_1, \dots, x_m, y_1, \dots, y_m$, the matrix representing $\Lambda$ takes the form of $m$ blocks of $2 \times 2$ matrices of the form

$$\lambda J := \begin{pmatrix} 0 & -\lambda \\ \lambda & 0 \end{pmatrix}. \tag{111}$$

In this way, we find that there is a basis for $\mathbb{R}^n$ relative to which $\Lambda$ has matrix

$$\begin{pmatrix} 0 & & & \\ & \lambda_1 J_{m_1} & & \\ & & \ddots & \\ & & & \lambda_k J_{m_k} \end{pmatrix}, \tag{112}$$

where $m_0 = \dim \ker \Lambda$ is the size of the first block, $m_j$ is the multiplicity of $i\lambda_j$ and $J_m$ is a $2m \times 2m$ matrix consisting of $m$ blocks of $2 \times 2$ matrices which are either

$$J = \begin{pmatrix} 0 & -1 \\ 1 & 0 \end{pmatrix} \qquad \text{or} \qquad -J = \begin{pmatrix} 0 & 1 \\ -1 & 0 \end{pmatrix}. \tag{113}$$

It is clear that the transformation taking $\Lambda$ to the above matrix is orthogonal, but perhaps not special orthogonal. Indeed, the matrices $\pm J$ are related by an orientation reversing transformation of the plane, and hence since we are only allowed to conjugate by $SO(n)$, we cannot simply make all blocks be $J$, say. This problem only arises for $n$ even, since if $n$ is odd, then $\Lambda$ has kernel and we can indeed use the freedom to multiply one of the basis elements in the kernel by $-1$ to ensure that the orthogonal matrix which conjugates $\Lambda$ to a normal form where all $2 \times 2$ blocks are of the form $J$, has determinant 1.

As an example, let us list the possible coadjoint orbits of $SO(4)$:

- If $\Lambda = 0$, we get a point-like orbit with stabiliser $SO(4)$.

- If $\dim \ker \Lambda = 2$, then we can bring $\Lambda$ to the form

$$\begin{pmatrix} 0 & 0 & 0 & 0 \\ 0 & 0 & 0 & 0 \\ 0 & 0 & 0 & -\lambda \\ 0 & 0 & \lambda & 0 \end{pmatrix}, \tag{114}$$

for some real $\lambda \neq 0$. We can arrange for $\lambda > 0$ without loss of generality, since, e.g.,

$$\begin{pmatrix} 1 & 0 & 0 & 0 \\ 0 & -1 & 0 & 0 \\ 0 & 0 & 0 & 1 \\ 0 & 0 & 1 & 0 \end{pmatrix} \in SO(4), \tag{115}$$

and conjugating by that matrix changes $\lambda$ to $-\lambda$. The stabiliser in this case is $S(O(2) \times U(1))$.

- If $\dim \ker \Lambda = 0$, we have two possibilities:

  - if the eigenvalue $\lambda$ has multiplicity 2, then we have two kinds of orbits corresponding to

  $$\begin{pmatrix} \lambda J & 0 \\ 0 & \lambda J \end{pmatrix} \quad \text{or} \quad \begin{pmatrix} \lambda J & 0 \\ 0 & -\lambda J \end{pmatrix}. \tag{116}$$

  Those two matrices are conjugate in $O(4)$, but not in $SO(4)$. The stabiliser of these coadjoint orbits is $U(2) \subset SO(4)$.

  - if the two eigenvalues $\lambda_1, \lambda_2$ are different, then we can assume without loss of generality that $\lambda_1 \leqslant \lambda_2$, and we can bring $\Lambda$ to one of the following two matrices

  $$\begin{pmatrix} \lambda_1 J & 0 \\ 0 & \pm\lambda_2 J \end{pmatrix}. \tag{117}$$

  The stabiliser in this case is $S(U(1) \times U(1))$.

Alternatively, we can argue as follows. Given any $\Lambda \in \mathfrak{so}(n)$, it is conjugate under $SO(n)$ to a matrix in a Cartan subalgebra, say:

$$\begin{pmatrix} 0 & \lambda_1 & & & & \\ -\lambda_1 & 0 & & & & \\ & & 0 & \lambda_2 & & \\ & & -\lambda_2 & 0 & & \\ & & & & \ddots & \\ & & & & & 0 \end{pmatrix}, \tag{118}$$

where the $\lambda_i$ are not necessarily distinct. This still leaves the action of the Weyl group. If $n = 2\ell + 1$ is odd, then the Weyl group is $\{\pm 1\}^\ell \rtimes S_\ell$: the symmetric group $S_\ell$ acts by permuting the $\lambda_i$ and $\{\pm 1\}^\ell$ acts by changing the signs of the $\lambda_i$. We can therefore arrange them so that $0 \leqslant \lambda_1 \leqslant \lambda_2 \leqslant \cdots \leqslant \lambda_\ell$. If $n = 2\ell$ is even, then the Weyl group is the index-2 subgroup of $\{\pm 1\}^\ell \rtimes S_\ell$ consisting of an *even* number of sign changes, equivalently, it is the subgroup $K_\ell \rtimes S_\ell$, where $K_\ell$ is the kernel of the group homomorphism $\{\pm 1\}^\ell \to \{\pm 1\}$ sending $(\sigma_1, \ldots, \sigma_\ell) \mapsto \sigma_1 \ldots \sigma_\ell$.

In the case of $SO(4)$ we can bring any $\Lambda \in \mathfrak{so}(4)$ to

$$\begin{pmatrix} 0 & \lambda_1 & & \\ -\lambda_1 & 0 & & \\ & & 0 & \lambda_2 \\ & & -\lambda_2 & 0 \end{pmatrix}, \tag{119}$$

and the Weyl group is the Klein *Viergruppe* acting on $(\lambda_1, \lambda_2)$ by permuting them $(\lambda_1, \lambda_2) \mapsto (\lambda_2, \lambda_1)$ or changing *both* their signs $(\lambda_1, \lambda_2) \mapsto (-\lambda_1, -\lambda_2)$. The moduli space of coadjoint orbits is then the wedge $-\lambda_2 \leqslant \lambda_1 \leqslant \lambda_2$, which is illustrated in Figure 2. The generic stabiliser is $S(U(1) \times U(1))$. If $\lambda_1 = 0$, but $\lambda_2 \neq 0$, then the stabiliser is $S(O(2) \times U(1))$. If $\lambda_1 = \pm\lambda_2 \neq 0$, then the stabiliser is enhanced to $U(2)$, and if $\lambda_1 = \lambda_2 = 0$, it is all of $SO(4)$.

# D   Coadjoint orbits of the Lorentz group

In this appendix we describe the coadjoint orbits of the (proper, orthochronous) Lorentz group; that is, the identity component $SO(n, 1)_0$ of the $(n + 1)$-dimensional Lorentz group, for $n \geqslant 2$. Since for those values of $n$, $\mathfrak{so}(n, 1)$ is semisimple, the musical isomorphisms associated to

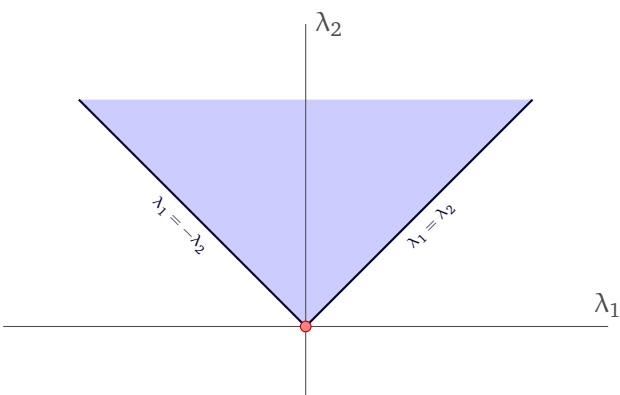

Figure 2: Moduli space of coadjoint orbits of SO(4).

the Killing form, which are $\mathfrak{so}(n, 1)$-equivariant, set up an equivalence between the adjoint and coadjoint representations and hence between the adjoint and coadjoint orbits under the identity component $SO(n, 1)_0$.

As explained, for example, in a different context in [56, Section 2], every $\Lambda \in \mathfrak{so}(p, q)$ can be brought to a normal form under the adjoint action of $SO(p, q)$. In that normal form, $\Lambda$ is block diagonal where the relevant elementary blocks for $(p, q) = (n, 1)$ are given in Table 8, together with their signature $(p, q)$. To obtain the normal forms for $\Lambda \in \mathfrak{so}(n, 1)$, we need to build all block diagonal matrices of signature $(n, 1)$ using the elementary blocks and then check whether there is a further identification of the parameters under the adjoint action.

Table 8: Elementary blocks for lorentzian signature. The parameters $\varphi, \beta$ are nonzero real numbers, but depending on $n$, we may restrict their sign.

| Signature $(p, q)$ | Block in $\mathfrak{so}(p, q)$ |
|---|---|
| $(1, 0)$ | $[0]$ |
| $(0, 1)$ | $[0]$ |
| $(2, 0)$ | $\begin{bmatrix} 0 & \varphi \\ -\varphi & 0 \end{bmatrix}$ |
| $(1, 1)$ | $\begin{bmatrix} 0 & \beta \\ \beta & 0 \end{bmatrix}$ |
| $(2, 1)$ | $\begin{bmatrix} 0 & 1 & 0 \\ 1 & 0 & \pm 1 \\ 0 & \mp 1 & 0 \end{bmatrix}$ |

For example, if $n = 2$, then we can build a block-diagonal matrix of signature $(2, 1)$ from elementary blocks as follows:

- $(0, 1) \oplus (1, 0) \oplus (1, 0)$

- $(0, 1) \oplus (2, 0)$

- $(1, 1) \oplus (1, 0)$

- $(2, 1)$

This results in the following normal forms for matrices in $\mathfrak{so}(2,1)$:

$$
\begin{pmatrix} 0 & 0 & 0 \\ 0 & 0 & 0 \\ 0 & 0 & 0 \end{pmatrix}, \qquad
\begin{pmatrix} 0 & 1 & 0 \\ 1 & 0 & \pm 1 \\ 0 & \pm 1 & 0 \end{pmatrix}, \qquad
\begin{pmatrix} 0 & 0 & 0 \\ 0 & 0 & \varphi \\ 0 & -\varphi & 0 \end{pmatrix}, \qquad
\begin{pmatrix} 0 & \beta & 0 \\ \beta & 0 & 0 \\ 0 & 0 & 0 \end{pmatrix}. \tag{120}
$$

The two irreducible blocks cannot be related under the identity component $SO(2,1)_0$, and neither can the sign of $\varphi$ be changed. However, there is a proper orthochronous Lorentz transformation which changes the sign of $\beta$ in the last block, so that we can actually take $\beta > 0$. Because the Killing form for $\mathfrak{so}(2,1)$ has signature $(2,1)$, the coadjoint orbits of $SO(2,1)_0$ coincide with the orbits in a three-dimensional lorentzian vector $(V, \eta)$ space under the proper orthochronous Lorentz transformations: namely,

- the origin, corresponding to the zero matrix;

- the future and past deleted lightcones $\eta(v, v) = 0$, corresponding to the matrices $\begin{pmatrix} 0 & 1 & 0 \\ 1 & 0 & \pm 1 \\ 0 & \pm 1 & 0 \end{pmatrix}$;

- the upper and lower sheets of the two-sheeted hyperboloids $\eta(v, v) = -\varphi^2$, corresponding to the matrices $\begin{pmatrix} 0 & 0 & 0 \\ 0 & 0 & \varphi \\ 0 & -\varphi & 0 \end{pmatrix}$ for $\pm \varphi > 0$; and

- the one-sheeted hyperboloid $\eta(v, v) = \beta^2$, corresponding to the matrices $\begin{pmatrix} 0 & \beta & 0 \\ \beta & 0 & 0 \\ 0 & 0 & 0 \end{pmatrix}$, for $\beta > 0$.

This of course is completely elementary and is the lorentzian analogue of the decomposition of three-dimensional euclidean space under the group of rotations into the origin and the spheres of radius $r$ for $r > 0$.

To illustrate further, let us now consider $n = 3$ and discuss the coadjoint orbits of $SO(3,1)_0$. Using the elementary blocks in Table 8, we can build a block diagonal matrix of signature $(3,1)$ as follows:

- $(0,1) \oplus (1,0) \oplus (1,0) \oplus (1,0)$

- $(2,1) \oplus (1,0)$

- $(1,1) \oplus (2,0)$

- $(1,1) \oplus (1,0) \oplus (1,0)$

- $(0,1) \oplus (1,0) \oplus (2,0)$

This results in the following normal forms for matrices in $\mathfrak{so}(3,1)$:

$$
\begin{pmatrix} 0 & 1 & 0 & 0 \\ 1 & 0 & \pm 1 & 0 \\ 0 & \mp 1 & 0 & 0 \\ 0 & 0 & 0 & 0 \end{pmatrix}, \quad
\begin{pmatrix} 0 & \beta & 0 & 0 \\ \beta & 0 & 0 & 0 \\ 0 & 0 & 0 & \varphi \\ 0 & 0 & -\varphi & 0 \end{pmatrix}, \quad
\begin{pmatrix} 0 & \beta & 0 & 0 \\ \beta & 0 & 0 & 0 \\ 0 & 0 & 0 & 0 \\ 0 & 0 & 0 & 0 \end{pmatrix}, \quad
\begin{pmatrix} 0 & 0 & 0 & 0 \\ 0 & 0 & 0 & 0 \\ 0 & 0 & 0 & \varphi \\ 0 & 0 & -\varphi & 0 \end{pmatrix}, \tag{121}
$$

in addition to the zero matrix. Now conjugation with $SO(3,1)_0$ relates the versions of the first matrix with the different signs, so we need only consider one of them. Similarly, in the last two normal forms, we need only keep $\beta > 0$ and $\varphi > 0$. In the second normal form, conjugation changes the signs of $\beta$ and $\varphi$ simultaneously, hence we can take $\beta > 0$ and $\varphi$ nonzero but otherwise unconstrained.

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
