# Peer review of "Lifshitz symmetry: Lie algebras, spacetimes and particles"

_SciPost Physics, doi:SciPost Phys. 14, 035 (2023)_

## Round 2 · Referee Report · Anonymous (Referee 1) · 2022-8-4

Strengths

1- The paper addresses an original and timely question, namely what is the complete set of Lifschitz Lie algebras (as defined in the paper and naturally suggested by "standard" Lifschitz space-time symmetry), what are their central extensions, their homogeneous spaces, and their coadjoint orbits. 2- The paper is well written and seems to be correct and complete: a reader willing to repeat the necessary computations will find most of the needed material in the paper, up to prior works that are suitably cited.

Weaknesses

1- Very technical! The motivation comes from physics of course, but reading the paper sometimes feels like an algebraic exercise. I'm not familiar enough with the subject of Lifschitz symmetry to suggest anything concrete, but perhaps the authors could mention relevant physical examples when possible. This would connect the abstract algebra to specific physical situations.

[Examples: (1) The central extension in $\mathfrak{a}_6$ seems to be that of non-commuting magnetic translations, connecting the algebra to the physics of planar electrons in a strong magnetic field. Why not mention this? (2) How to think of the contractions listed in fig. 2? What is the interpretation of the "parameter that goes to zero" along each arrow? (3) What's the physical interpretation of the invariants specifying orbits between eqs. (6.18) and (6.19)? Mass and spin?]

Report

The paper addresses an original and timely issue in mathematical physics. It seems to be correct and complete, and is very well written despite its technical subject matter, so I recommend publication in SciPost with only minor modifications (see below).

I stress that most of these modifications are optional, as the paper would deserve to be published even without them - it's just that implementing them would strengthen the paper.

Requested changes

1-[optional] Related to the "weakness" mentioned above, the paper would be improved by relating the various mathematical objects it studies (algebras, extensions, manifolds, orbits) to concrete physical systems and observables. I don't mean this in the sense of, say, adding more references or motivation in the introduction (the introduction is already very well written and the motivations are clear). Rather, I mean that adding such notes "in passing" in the main text would make the whole discussion more widely accessible.

2-[optional] A comment for the conclusion: what would be the irreducible unitary representations of Lifschitz Lie groups? Can they be obtained by quantizing their coadjoint orbits? How are they related to standard "one-particle states" (relativistic or not)? These questions are naturally part of another paper (a follow-up?), and I was surprised not to see them mentioned in the conclusion, which otherwise correctly points out that it would be natural to add boost generators to Lifschitz algebras.

3- Minor clarifications needed: (a) At the end of section 1, a sentence reads as "(...) and the orbits of the one-parameter subgroup $\langle\xi_D\rangle$ generated by $\xi_D$ hit the hypersurface $r=0$ at precisely one point". If I understand this sentence correctly, I believe it would be more correct to write it as follows: "(...) and each integral curve of $\xi_D$ hits the hypersurface $r=0$ at precisely one point" (b) Above eq. (3.2), the text reads "the $\mathfrak{so}(d)$-equivariance of the brackets...". Does "$\mathfrak{so}(d)$-equivariance" means the same as "validity of Jacobi identities involving the generators $J_{ab}$"? If yes, it might be simpler to just write that.

4- Minor typos: (a) Below eq. (2.1): "there are seven class[es] of" (b) At the start of section 2.2: "Lie algebras [in] Table 1" (c) Section 4.2.8: "in Table[s] 3 and 4" (d) Section 4.2.8: The text reads as "in the case it acts effectively or Q in case the action is not", which should probably be replaced by "in the case it acts effectively or Q in case it does not" (e) Section 6.2.8: "regardless [of] the causal type" (f) Section 6.2.9: "but [it] might be interesting to study [them] in the future" ? (g) Section 7: "Another consequence of our classification [is] that"

  • validity: high
  • significance: good
  • originality: high
  • clarity: high
  • formatting: perfect
  • grammar: excellent

Author:  Stefan Prohazka  on 2022-10-06  [id 2884]

(in reply to Report 1 on 2022-08-04)
Category:
remark
correction

We thank the referee for the useful feedback and comments.

Requested changes:

  1. We have made two changes to adress this point:

    • We have added Section 4.3 which provides additional explanations of the limits.
    • We have added a discussion of the invariant of the coadjoint orbit.
  2. We have added the point ``Unitary irreducible representations'' to the discussion that highlights this possibility. We were hesitant because the concept of "particle" is less clear cut for conformal field theories and we are not aware of the state-of-the art for Lifshitz-like theories. But we completely agree with that this is an interesting natural question that deserves further study.

  3. (a) We have followed the referee's suggestion and simplified the statement to refer to the integral curves of $\xi_D$, but of course since $\xi_D$ is a fundamental vector field of a group action, its integral curves are the orbits of the one-parameter subgroup generated by $D$. (b) It is indeed the case that the Jacobi identity with $J_{ab}$ is precisely the $so(d)$-equivariance of the brackets. It's a matter of taste which statement is preferred and we have followed the suggestion of the referee.

  4. We have corrected the typos.

---

## Round 2 · Referee Report · Anonymous (Referee 2) · 2022-9-19

Strengths

1). Very solid work.
2). Clearly written and presented.
3). Results are easy to look up due to summary of results and use of tables.

Weaknesses

1). There is a mismatch between the introduction and conclusion sections, and the rest of the paper. The former are physics motivated, and the latter is pure maths.

Report

The paper defines and classifies Lifshitz algebras, and its Klein pairs that give rise to homogeneous spacetimes. Furthermore, the paper studies central extensions and co-adjoint orbits of these Lifshitz groups.

The results are entirely mathematical. No attempt at applying this to any physical system is given (not a judgement, just an observation).

I do not really see the connection with fracton theories as mentioned in the conclusions. In the fracton case the algebra has a dipole charge that does not commute with momenta, while e.g. in table 4 there is a scalar charge that does not commute with the Hamiltonian. It would have been nice if the authors would have presented a simple field theory with such a symmetry, so that it is a bit better motivated. At the level this is presented at, it is not clear that this has any physical interest.

There are two comments in the introduction that I believe are not correct, and that I would like the authors to address:

1). It is not in general true that effective theories in condensed matter physics away from critical points are Lorentz invariant.

2). The second incorrect comment is that in a holographic duality (in the large N limit) the strongly coupled regime of the QFT is dual to weakly coupled gravity. The latter should be classical gravity, but this is often not weakly coupled. The confusion is probably with weakly coupled strings on AdS, but in an AdS/CMT setup it is usually just classical gravity that is being considered.

Requested changes

Address points 1 and 2 listed in the report.

  • validity: top
  • significance: good
  • originality: ok
  • clarity: top
  • formatting: perfect
  • grammar: perfect

Author:  Stefan Prohazka  on 2022-10-06  [id 2885]

(in reply to Report 2 on 2022-09-19)
Category:
remark
reply to objection
correction

We thank the referee for the useful feedback and comments.

Concerning the paragraph on the connection with fracton theories: We hope that the connection to fracton-like symmetries on the level of the spacetimes is somehow clear, but we agree with the open question concerning the field theory realisation and the physical relevance. In the conclusion we have therefore changed the last paragraph of "Exotic spacetime symmetries" (we have partially merged it into "Generalisation") and added a new point "Exotic aristotelian theories with scalar charge" with the following content:

"By showing that these symmetries exist and can be consistently realised we have provided the first nontrivial step for the construction of exotic aristotelian theories with scalar charge. However, to further clarify their physical relevance and the relation to fracton-like theories it would be interesting to construct field theories that realize these symmetries (e.g., following~\cite{Hirono:2022dci})."

** Requested changes:**

  1. We agree that effective theories in condensed matter physics are not necessarily Lorentz invariant. We have clarified:

    "At these points, Lorentz invariance is broken, and we observe an anisotropic scaling between time and space, $t \rightarrow \lambda^z t \quad \text{and} \quad x \rightarrow \lambda x,$ where $z \neq 1$. However, away from these critical points, we see Lorentz invariance restored, corresponding to $z=1$ above.'' to "For systems with Lifshitz invariance, Lorentz invariance is broken, and we observe an anisotropic scaling between time and space, $t \rightarrow \lambda^z t \quad \text{and} \quad x \rightarrow \lambda x,$ where $z \neq 1$. However, away from these critical points, there is the option that Lorentz invariance is restored, corresponding to $z=1$ above."

  2. We have deleted the offending "weakly-coupled" and left it unspecified to account for the wealth of options that AdS/CMT might present. We have changed: "underlying these systems and weakly-coupled gravitational theories" to "underlying these systems and gravitational theories".

---

## Round 3 · Referee Report · Anonymous (Referee 1) · 2022-10-7

Report

SciPost acceptance criteria are fully met -- I recommend publication.

---

## Round 3 · Referee Report · Anonymous (Referee 2) · 2022-10-26

Strengths

The strengths are the same as in my previous report.

1). Very solid work.
2). Clearly written and presented.
3). Results are easy to look up due to summary of results and use of tables.

Weaknesses

Also here nothing has changed. I my opinion there are no real weaknesses other than on the motivational front. It would have been nice with a bit more physical intuition or argumentation for why this classification is useful and how the results impact on certain physical problems.

Report

I am happy with the changes made by the authors and I recommend the paper for publication.

---

## Round 3 · Author Response

We thank the referees for their useful feedback and comments. We have taken it into consideration and we have made various changes as listed in the list of changes and the replies to the referees.

---

## Round 3 · List of Changes

Additionally to the changes that are described in the answers to the referees we have made the following change:

-) We corrected a mistake in L2 and L7 in Table 2. The changes in L7 required some explanation which we have added to Section 4.1.

---

## Editorial Decision

published